# Integrating Noise Pollution into Life Cycle Assessment: A Comparative Framework for Concrete and Timber Floor Construction

Rabaka Sultana [ID], Taslima Khanam *[ID] and Ahmad Rashedi

Faculty of Science and Technology, Charles Darwin University, Ellengowan Drive, Casuarina, NT 0810, Australia; rabaka.sultana@cdu.edu.au (R.S.); mabrur.rashedi@cdu.edu.au (A.R.)
* Correspondence: taslima.khanam@cdu.edu.au

## Abstract

Despite the well-documented health risks of noise pollution, its impact remains overlooked mainly in life cycle assessment (LCA). This study introduces a methodological innovation by integrating both traffic and construction noise into the LCA framework for concrete construction, providing a more holistic and realistic evaluation of environmental and health impacts. By combining building information modeling (BIM) with LCA, the method automates material quantification and assesses both environmental and noise-related health burdens. A key advancement is the inclusion of health-based indicators, such as annoyance and sleep disturbance, quantified through disability-adjusted life years (DALYs). Two scenarios are examined: (1) a comparative analysis of concrete versus timber flooring and (2) end-of-life options (reuse vs. landfill). The results reveal that concrete has up to 7.4 times greater environmental impact than timber, except in land use. When noise is included, its contribution ranges from 7–33% in low-density regions (Darwin) and 62–92% in high-density areas (NSW), underscoring the critical role of local context. Traffic noise emerged as the dominant source, while equipment-related noise was minimal (0.3–1.5% of total DALYs). Timber slightly reduced annoyance but showed similar sleep disturbance levels. Material reuse reduced midpoint environmental impacts by 67–99.78%. Sensitivity analysis confirmed that mitigation measures like double glazing can cut noise-related impacts by 2–10% in low-density settings and 31–45% in high-density settings, validating the robustness of this framework. Overall, this study establishes a foundation for integrating noise into LCA, supporting sustainable material choices, environmentally responsible construction, and health-centered policymaking, particularly in noise-sensitive urban development.

**Keywords:** life cycle assessment; concrete; timber; noise impact; exposure–response relationship; BIM

## 1. Introduction

### 1.1. Background

The world is experiencing critical environmental challenges, including climate change, resource depletion, pollution, and biodiversity loss. These challenges are enhanced by rapid population growth and urbanization, which drive significant development activities [1]. The construction industry is vital among the many sectors contributing to these issues [2–4]. It accounts for substantial energy consumption and material usage, often resulting in

environmental degradation [5]. Concrete, a primary material in construction, is widely used in residential, commercial, and industrial buildings. In Australia alone, the construction of more than 43,000 residential buildings in 2021 underscores the growing demand for concrete and associated environmental impacts [6].

Concrete production and its application in construction are resource-intensive processes that contribute significantly to environmental pollution [7]. Heavy machinery such as pavers, loaders, compressors, cranes, excavators, and transportation modes like dump trucks and concrete mixers amplify the environmental burden [8]. Beyond the well-documented carbon emissions, concrete-related activities generate noise pollution, an often-overlooked yet critical environmental issue [8]. Noise emissions occur throughout the life cycle of concrete, including raw material acquisition, production, transportation, construction, and end-of-life phases.

Noise pollution, characterized as unwanted or harmful sound, has far-reaching implications for human health and the environment [9–12]. High noise levels can result in physiological and psychological effects, including hearing impairment, cardiovascular disorders, sleep disturbances, and general annoyance [13,14]. In Europe, approximately 20–30% of the population is affected by noise-related health impacts [8,9]. In Australia (Adelaide), 28% of the population is also annoyed by noise pollution [15]. Occupational noise in construction is a significant concern, with sound levels often ranging from 80 to 130 decibels, exceeding the Occupational Safety and Health Administration (OSHA) limit of 90 decibels for an eight-hour workday. Construction workers, particularly equipment operators, carpenters, and plumbers, are disproportionately affected, with nearly 50% experiencing perceived hearing loss due to prolonged exposure [16]. Moreover, traffic noise—another significant contributor—frequently surpasses 55 decibels, affecting 40% of the European population during the day and 30% at night [17]. Daytime exposure leads to annoyance, while nighttime noise disrupts sleep, further exacerbating health issues [18]. In Australia, the median noise range is 78 decibels for traffic noise, which is also responsible for annoyance, sleep deprivation, etc. [19].

Despite its widespread impacts, noise pollution remains inadequately addressed in environmental impact assessments. Traditional environmental evaluation tools, including life cycle assessment (LCA), primarily focus on environmental factors such as carbon emissions, energy use, and resource depletion [20–23]. While there is growing recognition of acoustic comfort in sustainable building certifications like LEED (Leadership in Energy and Environmental Design) and BREEAM (Building Research Establishment Environmental Assessment Method), these systems lack robust frameworks for quantifying noise-related impacts [24]. Integrating noise into LCA can bridge this gap, offering a comprehensive assessment of the environmental effects. However, this integration is facing some challenges, including limited life cycle inventory (LCI) data for noise emissions, the complexity of modeling noise propagation, and the absence of standardized methods for incorporating noise impacts into LCA calculations [10,13,17,25–28].

Current research on noise pollution in the construction sector is fragmented. Studies have primarily focused on isolated noise sources, such as traffic or machinery, without considering their combined effects [18,29–31]. For instance, researchers have evaluated the noise impact of road traffic using propagation models like ISO 9613-1 ("Acoustics—Attenuation of sound during propagation outdoors—Part 1: Calculation of the absorption of sound by the atmosphere"). Still, these studies often neglect the complexities of non-circular noise propagation in highways because noise propagation can vary with vehicle type, road characteristics, and location [26]. Thus, these factors need to be considered to calculate the noise level. Similarly, machinery noise has been assessed in terms of its impact on human health, using indicators such as the number

of highly annoyed individuals and disability-adjusted life years (DALYs) for endpoint assessments [10,13,25,32,33]. However, there is a notable absence of studies integrating static (machinery) and mobile (traffic) noise sources into a unified LCA framework.

*1.2. Recent Literature Review*

Steen introduced the first noise-integrated LCA method, employing a monetization approach [18]. It estimated the cost of traffic noise on human health, categorizing noise above 65 dB as a nuisance, particularly during rush hours (assumed to be 4 h daily). This method also accounted for fuel consumption (1 kg/10 km) as an additional environmental impact. However, it had the following significant limitations: exposed population: assumed to be 25% of the global population, leading to overestimations; health impacts: focused only on midpoint impacts (e.g., exposed individuals) and ignored DALYs, a key measure of health damage.

Muller-Wenk enhanced noise integration by introducing a framework based on chemical emission analysis [17]. This model included four modules: fate analysis, exposure analysis, effect analysis, and damage analysis. It addressed both midpoint impacts, such as communication disturbance and sleep disturbance, and endpoint impacts using DALYs. While the framework was significant, it had limitations: it considered only light vehicles (cars, vans) and heavy vehicles (trucks, buses) as case studies. It used the Zurich data extrapolation method for exposed population estimates, limiting its applicability to specific regions. Later models introduced a linear noise growth assumption, which proved unrealistic since noise typically increases logarithmically. This led to overestimated noise levels.

A breakthrough came with Miedema and Oudshoorn, who introduced a polynomial dose–response curve to estimate highly annoyed individuals [34]. However, this method excluded endpoint impacts. Ongel later applied it to assess health outcomes such as annoyance, acute myocardial infarction, and sleep disturbance [35].

The most recent framework, developed by Meyer et al., introduced a five-step process: define the noise characterization model; select a reference flow for noise; choose midpoint indicators (e.g., highly annoyed and highly sleep-deprived individuals); define the modeling perspective; and compute the characterization factor (CF) based on population density and sound energy density [27]. This framework represents a significant step forward, emphasizing that CF values vary depending on contextual factors.

Besides applying the LCA method to traffic noise, some researchers assessed the occupational construction noise using the LCA method [36]. Occupational noise can be generated in the industry or on a construction site. Occupational noise impacts include cardiovascular risks, hearing loss, and psychosocial stress [37]. Despite evidence of these effects, few studies integrate them into LCA. In one study, the authors analyzed the noise emission in the same way as air pollution. Air pollution impact assessment quantifies impacts based on energy inputs. However, this approach oversimplifies the distinct characteristics of noise [38]. Like highly annoyed workers, other researchers often use polynomial dose–response approaches to estimate the effects. Since workers are already in industrial settings, sleep disturbance is typically excluded. Other studies analyzed noise in industries like factories and cogeneration plants, applying a four-step process: noise propagation, exposure, effect, and damage analysis [18]. However, many assessments remain geographically restricted or lack integration with LCA.

While traffic noise and construction machinery noise have been individually recognized for their environmental and health impacts, they are rarely assessed together in an integrated framework. In real-world construction scenarios, these noise sources co-occur, cumulatively affecting both nearby residents and on-site workers. Traffic noise, primarily

generated during material transportation phases, contributes significantly to community annoyance and sleep disturbance, especially in urban areas with high population density. Meanwhile, machinery noise, dominant during on-site construction activities, poses serious occupational health risks, including hearing loss and cardiovascular disorders among workers. Ignoring either source leads to an incomplete and underestimated assessment of the total noise-related burden. Furthermore, existing environmental evaluation tools and policies tend to address these sources separately, resulting in fragmented and less effective mitigation strategies. By integrating both mobile (traffic) and stationary (machinery) noise within the LCA framework, this study offers a more comprehensive and accurate method for quantifying the environmental and health impacts of concrete construction across its entire life cycle.

### 1.3. Aim

This study aims to develop a comprehensive and unified framework for integrating noise pollution into the life cycle assessment (LCA) of concrete construction. Unlike existing approaches that consider either traffic or construction noise in isolation, this research proposes a dual-source model that simultaneously incorporates mobile (traffic-related) and stationary (construction machinery) noise emissions across the entire concrete life cycle—from raw material extraction to end-of-life disposal.

### 1.4. Present Study

In this research, the authors assessed the impact of the concrete construction (foundation) of a single-storied residential building. Two types of impact will be assessed, material impact and noise impact. Material impact will be assessed by the ReCiPe 2016 method. In this method, both midpoint and endpoint impacts will be assessed. To achieve this, building information modeling (BIM) tools, such as Revit, are employed to calculate the concrete quantities required for residential buildings. Building information modeling integration enhances the efficiency and accuracy of LCA calculations by automating material quantification and streamlining data management [39–41]. In this proposed framework, noise impacts are then assessed using a two-tiered approach: midpoint indicators, such as the number of highly annoyed individuals and sleep-deprived populations, and endpoint indicators, expressed as DALYs, to capture the broader health consequences of noise pollution. In this study, the impact of noise on timber material is also assessed and compared with that of concrete construction. Additionally, recycling and landfill scenarios of those materials are compared and evaluated to measure sustainability including noise impact. A sensitivity analysis was conducted to determine how variations in key parameters could affect the results, ensuring that the findings were robust under different assumptions and conditions.

The structure of this paper is as follows: Section 2 describes the noise impact assessment development and details the methodology for noise-related LCA; Section 3 presents the results and discussion; Section 4 highlights limitations; and Section 5 concludes with recommendations for future research.

## 2. Methodology

The standardization of the LCA methodology, including its principles and requirements, is defined by ISO 14040 and ISO 14044 [24]. The LCA process consists of four main stages: goal and scope definition, life cycle inventory (LCI), life cycle impact assessment (LCIA), and interpretation.

### 2.1. Goal and Scope Definition

This research aims to assess the adverse effects of noise generated during concrete construction on human health. The study uses LCA to evaluate the environmental impact

of concrete production, steel reinforcement, construction work, maintenance, repair, and disposal with a functional unit of 174 m$^2$ of concrete flooring (Figure 1). This house is a single-storied four-bedroom residential building. In addition, a comparative LCA analysis was conducted on timber floors of the same building. It is assumed that the load coming from the sheet roof, brick veneer wall, timber structural frame (wall and roof), and others is the same for both types of floor. The lifespan of this floor is assumed to be 100 years for both concrete and timber. The study meticulously considers two end-of-life scenarios for concrete and timber, such as landfill as well as reuse and recycling, demonstrating the comprehensiveness of our research.

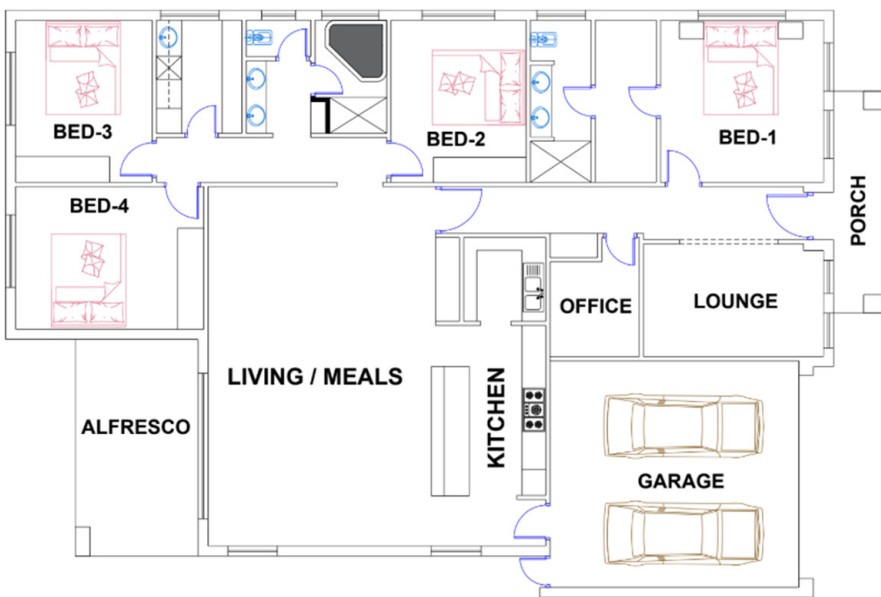

**Figure 1.** Floor plan of single-storied residential building.

### 2.2. Data Input/Inventory

Two data types were required to assess the environmental impact: material data and noise data. The authors gathered material data for concrete from its manufacturing stage to its end-of-life stage, along with noise data from equipment and transportation activities (Figure 2).

Figure 2 outlines the generation of materials and noise across six stages of a building's life cycle: material acquisition, manufacture, construction, use, maintenance, and end of life. During material acquisition, raw materials such as cement, sand, aggregates, and water were sourced for concrete production, while timber was obtained from forests. In the manufacture stage, concrete and reinforced concrete (integrating wire mesh) were prepared, and timber underwent treatment for use in floor foundations and floor finishes.

In the construction stage, building information modeling (BIM) was employed to streamline and improve the accuracy of material estimation and life cycle assessment inputs [42]. Using Autodesk Revit 2022, a 3D model of the building was developed based on 2D drawings. Project parameters were defined in detail, including material types, dimensions, and structural elements. BIM enabled the automated generation of a material takeoff schedule, providing a precise and consistent bill of quantities (BOQ) for concrete and timber floors (see Appendix A: Tables A1 and A2). These data were essential for quantifying material flows, energy use, and transportation impacts across life cycle stages. By leveraging BIM, the study ensured methodological consistency, minimized human error in quantity estimation, and strengthened the reproducibility of the LCA framework.

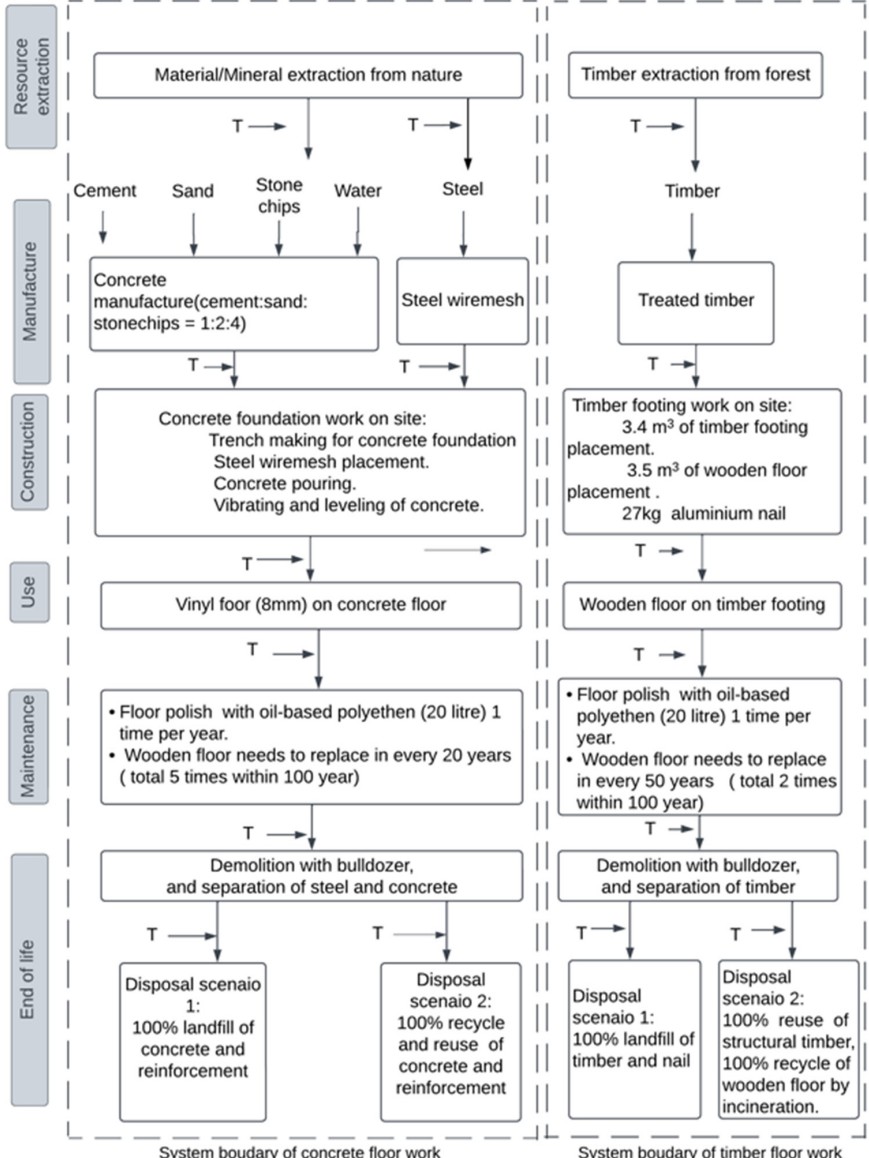

**Figure 2.** Flow chart of concrete and timber floor work. T = transportation. The arrow for transportation is indicates that transportation is required for this specific work.

Material consumption was projected over a 100-year building lifespan in the use and maintenance stages, with adjustments made based on repair and replacement frequencies. The final BOQ, incorporating equipment, transportation, and energy data, is summarized in Table A3 (see Appendix A). At the end-of-life stage, two waste management scenarios were evaluated—100% landfill disposal and 100% recycling of concrete and timber materials—to assess comparative environmental performance.

The research also incorporated noise data associated with all concrete and timber flooring activities, categorizing noise sources into mobile and stationary types. Mobile(/dynamic) sources include transportation noise while stationary sources encompass industrial and construction noise. It was assumed that one heavy vehicle and one light vehicle were used for transportation for mobile sources. Heavy vehicles, such as trucks, transport minerals from sites to industries, deliver concrete to construction sites, carry materials during maintenance, and transport demolition waste to landfill areas. Light vehicles, such as small cars, transported construction workers during all project phases, including mining, industry, construction, maintenance, and end-of-life stages. Tables A4 and A5

(see Appendix A) provide detailed information on distances and travel times for these transportation activities (these data are assumed).

For stationary sources, noise data were collected for various equipment used during construction [37]. This information is collected from different research. Equipment such as earth-moving machines, bulldozers, cranes, and piling hammers were used during mineral acquisition for concrete ingredients. Tools like concrete mixers, compressors, compression tools, and compactors were employed for concrete pouring. Similarly, excavators, jackhammers, timber-lifting cranes, and chainsaws were used for timber work. All these equipment and machinery noise levels are listed in Table A4 (see Appendix A) [38,43].

### 2.3. Life Cycle Impact Assessment

In the life cycle inventory analysis, two distinct data types were collected: material data and noise data, to develop a noise-integrated LCA method. By using material data, environmental impact can be assessed by the conventional LCA method. Since existing LCA methods do not adequately integrate noise impacts, this research aims to propose a comprehensive framework to address this gap. The framework consists of four main components: fate analysis, exposure analysis, effect analysis, and damage analysis (Figure 3).

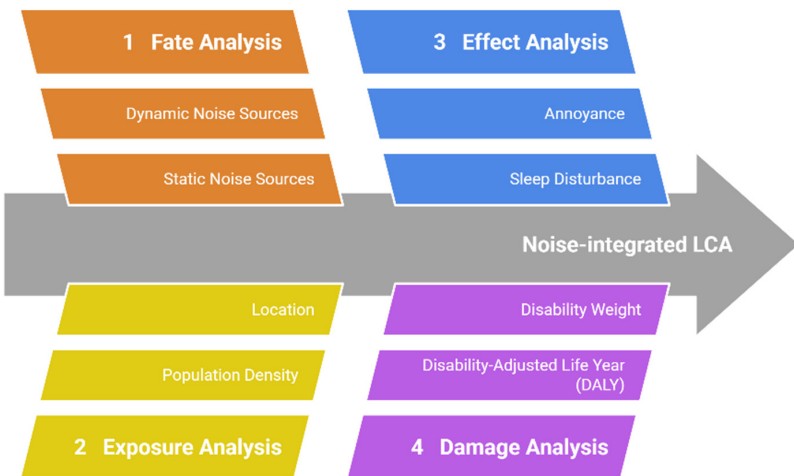

**Figure 3.** Cause–effect chain of noise propagation.

### 2.3.1. Fate Analysis

Fate analysis calculates the total noise generated from static and mobile sources. Static noise can also be expressed as stationary noise and mobile noise can be expressed as dynamic noise. This analysis also considers the specific propagation characteristics. Once the noise emission and propagation are accurately modeled, the study will adopt the methods for exposure analysis, effect analysis, and damage analysis developed by Muller et al. and Maidemaa et al. to establish a comprehensive noise evaluation approach tailored for construction work [16,17]. By integrating these components, the proposed framework aims to incorporate noise impact assessments seamlessly into the LCA process.

The following sections will explain each framework component, outlining how noise data are systematically incorporated into the LCA methodology.

Accurately calculating noise levels requires accounting for all significant noise sources, both static and dynamic, as each contributes uniquely to the overall noise impact. In this study, the noise emission and propagation model is developed following the European Directive 2002/49/EC, which considers A-weighted noise (the noise frequency range humans can hear) distribution in both spatial and temporal dimensions [44].

Static noise sources, such as equipment and machinery, are critical in industrial and construction settings. Their noise propagation is influenced by environmental factors like

location (urban, rural, or industrial) and conditions (rain, temperature, humidity) [45]. Dynamic noise sources, including cars and trucks, introduce additional complexities due to their mobility and dependence on traffic characteristics like volume, speed, and vehicle type.

Spatial distribution examines how noise propagates based on these environmental factors, ensuring that variations in location and environmental conditions are captured accurately. On the other hand, temporal variation highlights how noise levels fluctuate during different time zones (day, evening, and night), which is particularly relevant for dynamic sources like traffic. This study's static and dynamic sources are essential to ensure a holistic noise assessment. Traffic noise is especially significant as it depends on spatial and temporal variations, making it a critical contributor to the propagated noise levels.

By integrating all noise sources, this research evaluates the propagated noise at three key receivers:

1. Structural noise receiver: Individuals directly exposed to noise while using equipment, machinery, or transportation;
2. Airborne noise receiver: Individuals situated 50 feet away from the noise source, capturing the impact of noise propagation;
3. Indoor noise receiver: Occupants inside residential buildings affected by transmitted noise.

This approach ensures no noise source is overlooked, providing a comprehensive framework for assessing noise impacts. The detailed steps for noise propagation analysis are explained in Appendices A.1–A.3 [11,12,33,46–58].

In this research (Appendix A.1), the total noise level due to mobile sources and static sources has been calculated and described as follows.

Total static noise, $LA_{eq}y$(static), can be expressed as follows:

$$LA_{eq}\ y\ (static) = 10log\{LA_{eq,1}\ y(structural\ noise) + LA_{eq,2}\ y\ (airborne\ noise\ inside\ the\ industry) \\ + LA_{eq,3}\ y(outdoor\ noise\ in\ industry)\} \tag{1}$$

where $LA_{eq\,1}\ y$ is the structural noise level due to equipment that generates in a factory or a construction site, $LA_{eq,2}\ y$ is the airborne noise level due to the equipment, and $LA_{eq,3}\ y$ is the outdoor noise level due to equipment.

Static noise has three categories of receivers based on their proximity and exposure. The primary receivers are the equipment operators who experience structural noise through direct contact with the machinery. Secondary receivers include workers nearby exposed to airborne noise transmitted through the surrounding air. Lastly, tertiary receivers consist of individuals working outside the industry or residing within buildings where noise levels are significantly reduced due to the attenuation effect of exterior walls.

Transportation noise also has three receivers, similar to equipment noise. Here, structural noise receivers are those inside the vehicle. Airborne noise receivers are pedestrians and indoor noise receivers who live inside the building.

Total transportation noise, $LA_{eq}m$(mobile), will be as follows:

$$LA_{eq}\ m\ (mobile) = 10log\{LA_{eq,1}\ m(structural\ noise) + LA_{eq,2}\ m(airborne\ noise) \\ + LA_{eq,3}\ m(indoor\ noise\ of\ residential\ house)\} \tag{2}$$

where $LA_{eq,1}\ m$ is the structural noise level of the vehicle, $LA_{eq,2}\ m$ is the airborne noise level for pedestrians, and $LA_{eq\,3,}\ m$ is the indoor noise level.

The total $LA_{eq}$ (total) noise levels from construction noise sources can be combined using the following equations.

$$LA_{,total} = 10log\left(LA_{eq\ y\ (static)} + LA_{eq\ m(mobile)}\right) \tag{3}$$

where $LA_{eq}y$(static) is static noise and $LA_{eq}m$(mobile) is transportation noise.

### 2.3.2. Exposure Analysis

After analyzing the propagation model, exposure analysis is the next step. Population density is a critical factor in exposure analysis. Two locations in Australia are analyzed: Darwin (Casuarina to Gray) and NSW (Silverwater to Paddington) (Figure 4). The drawn lines in the figure indicate the regions beside those lines. Darwin represents a rural area with a low population density, while NSW is an urban area with a high population density.

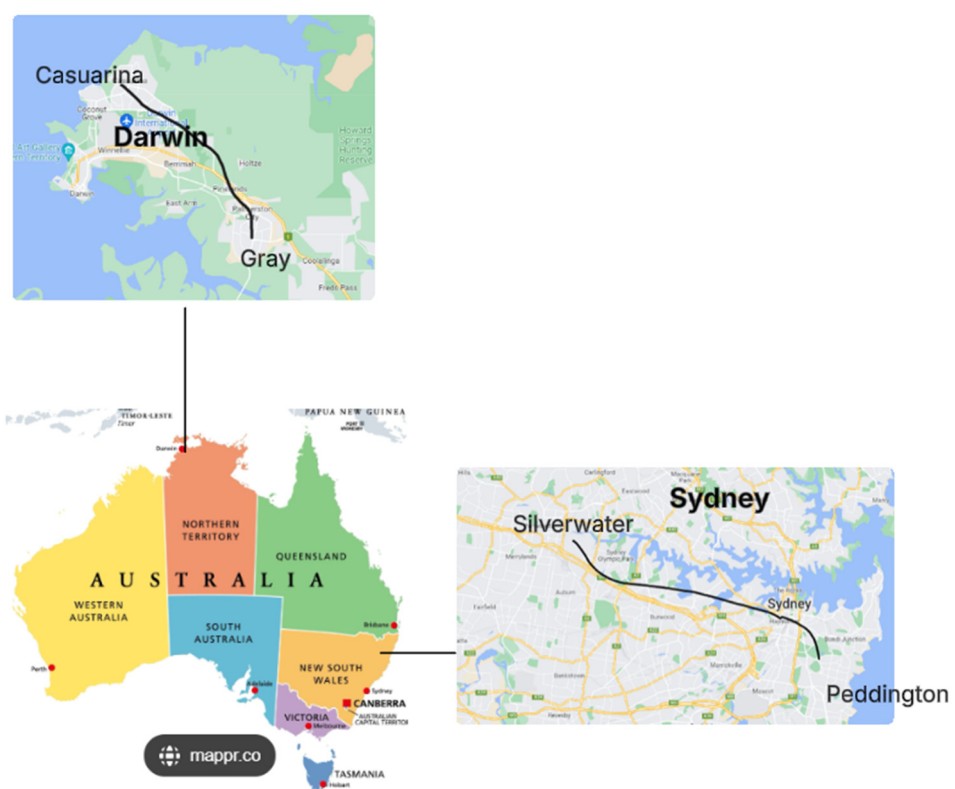

**Figure 4.** Two locations (Darwin and Sydney) in Australia selected for this case study. The horizontile line showing the location of NSW and its blow-up area. The line in the Sydney area indicates the road that start from Silberwater to Peddington. The vertical line from top of Australia map indicates the location of Darwin and its blow-up area. The line in Darwin shows the road that starts from Casuarina to Gray.

Noise pollution from concrete work is assessed in three key areas: the production site, the road transportation route, and the construction site. Noise generated at the production and construction sites has localized impacts, primarily affecting the immediate vicinity. The extent of exposure to noise pollution depends on the area's population density, which determines the number of inhabitants affected. This study assumes that 10% of the population in each region is exposed to noise pollution. This percentage of the population (10–20%) was determined based on similar research (10–20%) [18]. Thus, the exposed population of that zone was calculated by the following:

$$\text{Exposed population} = \text{population of that area} \times \text{specific area share} \times \text{noise level generated from the noise source.} \quad (4)$$

### 2.3.3. Effect Analysis

After the exposure analysis, the effect of noise needs to be calculated. Noise-induced health impacts include annoyance, sleep disturbance, cognitive effects, and other health-related issues [59]. The authors followed the known practice of calculating the midpoint impact, such as highly annoyed and sleep-deprived people. The authors used the polynomial approximation method for static and dynamic noise [18,60].

The highly sleep-deprived people (%HSDP) who are exposed to a sound pressure level at night (L$_{night}$) ranging from 45–65 decibels are expressed as the following equation [13]:

$$\%HSDP = 20.8 - 0.01486L_{night}^2 - 1.05L_{night} \tag{5}$$

L$_{night}$ is the sound pressure level measured at night, specifically from 23:00 to 07:00 a.m., which are sleep-sensitive hours.

The annoyance impact can include stress-related psychosocial symptoms, such as anger, disappointment, withdrawal, depression, anxiety, distraction, and agitation [61]. Physiological distress hampers mental health and well-being. Prolonged noise exposure can reduce attention and focus on work.

The percentage of highly annoyed people (%HA) who are exposed to LA$_{eq}$ between 45–75 decibels can be expressed as

$$\%HA = 0.5118\left(LA_{eq} - 42\right) - 0.01436\left(LA_{eq} - 42\right)^2 + 0.0009868\left(LA_{eq} - 42\right)^3 \tag{6}$$

Here, LA$_{eq}$ is the total noise calculated using Equation (3).

To calculate the number of highly annoyed people and highly sleep-deprived people, the exposed population needs to be multiplied by the percentage of highly annoyed/sleep-deprived people. Those are the midpoint impacts of noise.

### 2.3.4. Damage Analysis

After the effect analysis (midpoint impact), the damage analysis is the final stage that needs to be assessed for the LCA method. The human health impact for damage analysis can be expressed as disability-adjusted life years (DALYs) combined with year loss due to disability (YLD) and year loss due to life lost (YLL). To calculate the DALYs, the WHO suggested the disability weight of annoyance and sleep deprivation, such as 0.033 and 0.055 for annoyance and sleep deprivation, respectively [17,18]. Some researchers use the DALY values 0.02 and 0.07 for annoyance and sleep deprivation, respectively [33]. However, some researchers use nearly similar values (0.01–0.0175) for annoyance and sleep deprivation [30].

## 3. Result and Discussion

The noise-integrated life cycle assessment (LCA) method presented in this study was used to analyze the environmental impact of concrete construction. The assessment covers the full life cycle of concrete flooring, including material acquisition, construction, use, and end-of-life phases, each contributing to the overall environmental burden. The ReCiPe 2016 method was applied using SimaPro 9.4.0.2 software, with life cycle inventory data sourced from the Ecoinvent 3.8 database to ensure reliability. Additionally, this research integrates noise impacts associated with both concrete and timber construction activities. Two types of impact assessment methods—midpoint and endpoint—were used to evaluate the environmental impacts of concrete and timber floors, including their materials and related transportation. To assess the circular economy impact, two scenarios such as landfill vs. recycling were implemented, incorporating reduced transportation distances.

### 3.1. Midpoint Impact Assessment

#### 3.1.1. Impact of Concrete and Timber Floors (Material and Associated Traffic)

In this study, midpoint impact assessment was selected to provide a detailed, category-specific understanding of environmental burdens associated with construction activities. Midpoint indicators are grouped into two main categories: (1) traditional environmental indicators—such as global warming, stratospheric ozone depletion, ionic radiation, ozone

formation (human health and terrestrial ecosystem), fine particulate matter formation, terrestrial acidification, freshwater and marine eutrophication, ecotoxicity, toxicity, land use, resource scarcity, and water consumption—and (2) noise-specific health indicators, namely highly annoyed people (HAP) and highly sleep-deprived people (HSDP) [24]. The traditional indicators reflect impacts caused by emissions and resource consumption throughout the construction life cycle, such as raw material extraction, transportation, manufacturing, and energy use. These do not include noise emissions directly. In this study, however, noise impacts are integrated into the LCA framework through separate noise-specific health indicators (HAP and HSDP), thus addressing a critical limitation of conventional LCAs that overlook acoustic pollution.

It is important to note that no direct mechanistic linkage exists between noise and some midpoint categories, like terrestrial acidification or fossil resource scarcity. These categories are influenced primarily by material and energy flows, not acoustic emissions. However, noise impacts are included in the same LCA framework to offer a complementary perspective, comprehensively addressing environmental degradation and human health effects. Thus, the study presents both groups of indicators to emphasize the multi-dimensional nature of construction sustainability assessment (Appendix A.2).

Figure 5 compares the midpoint impacts of concrete and timber floors in Darwin and NSW. The values shown are relative percentages, where the highest impact value in each environmental category is scaled to 100%, and others are expressed proportionally (see Appendix A: Table A6). This visualization is intended for comparative purposes and does not represent methodological normalization (e.g., ReCiPe normalization to person-equivalents). Noise indicators—highly annoyed people (HAP) and highly sleep-deprived people (HSDP)—are based on exposure–response models and population data, presented here for reference but not normalized using LCA methods due to their different calculation basis. A separate breakdown of noise effects is provided in Figure 6.

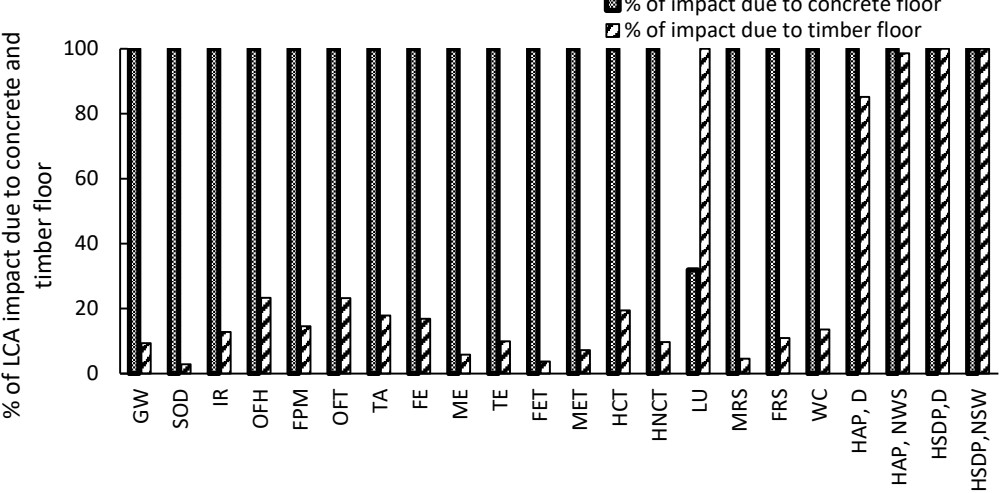

Noise integrated mid-point impact comparison between concrete and timber floor.

**Figure 5.** Noise-integrated midpoint impacts of concrete and timber floor in Darwin and NSW. GW = global warming; SOD = stratospheric ozone depletion; IR = ionic radiation; OFH = ozone formation, human health; FPM = fine particulate matter formation; OFT = ozone formation, terrestrial ecosystems; TA = terrestrial acidification; FE = freshwater eutrophication; ME = marine eutrophication; TE = terrestrial ecotoxicity; FET = freshwater ecotoxicity; MET = marine ecotoxicity; HCT = human carcinogenic toxicity; HNCT = human non-carcinogenic toxicity; LU = land use; MRS = mineral resource scarcity; FRS = fossil resource scarcity; WC = water consumption; HAP, D = highly annoyed people in Darwin; HAP, NSW = highly annoyed people in NSW; HSDP, D = highly sleep-deprived people in Darwin; HSDP, NSW = highly sleep-deprived people in NSW.

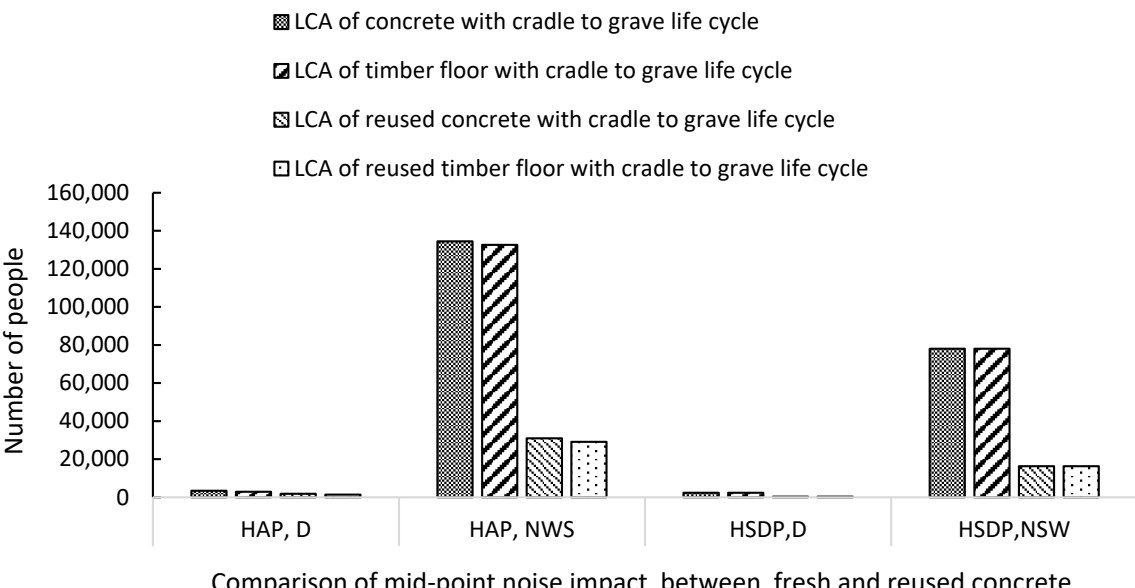

**Figure 6.** Midpoint noise impact of fresh and reused concrete and timber material. HAP, D = highly annoyed people in Darwin; HAP, NSW = highly annoyed people in NSW; HSDP, D = highly sleep-deprived people in Darwin; HSDP, NSW = highly sleep-deprived people in NSW.

In the conventional LCA method, the results show that concrete has a significantly higher impact than timber flooring, except for land use (Figure 5). For the noise impact indicator, annoyance is 14% and 1.4% less in Darwin for concrete flooring than for timber flooring (Figure 5). However, for sleep deprivation impact, both materials have the same effect in Darwin and NSW.

Figure 6 indicates that the total number of annoyed people for concrete floor work in Darwin and NSW are 3411 (for traffic, 1971, and for static, 1439) and 134,405 (for traffic, 129,288, and for static, 5117), respectively. The total number of annoyed people for timber floor work in Darwin and NSW are 2906 (for traffic, 1971, and for static, 934) and 132,552 (for traffic, 129,288, and for static, 3264), respectively, a bit less than concrete floor work. The total sleep-deprived people for concrete and timber floor work in Darwin and NSW are 2323 and 78,027, respectively. The same truck and car have been used for concrete and timber delivery. As a result, the total number of HAP and HSDP are nearly identical for concrete and timber work in both locations. This explicit inclusion of noise-related health impacts allows for side-by-side comparison with material-based environmental categories, offering a holistic view of the total burden caused by construction processes.

In this study, the number of highly annoyed people (HAP) and highly sleep-deprived people (HSDP) was primarily estimated using a percentage-based method. This approach assumes a uniform average population density within the noise-affected area and is commonly used in early-stage life cycle assessments (LCA) when high-resolution population data are unavailable.

To assess the robustness of this method, a supplementary analysis was conducted using Geographic Information System (GIS)-based population estimates. After calculating the construction noise levels, QGIS software version 3.4.8 was used to spatially estimate the number of people exposed to noise. Population data were sourced from the WorldPop database and integrated into QGIS for spatial analysis and visualization. As illustrated in Figure 4, two road segments were analyzed—one located in Darwin and the other in New South Wales (NSW)—to represent low- and high-populated areas, respectively. The estimated average population density in Darwin is approximately 17 persons/km [31]. For

NSW, average population density is assumed 82 persons/km which is same density of city area of Darwin.

The comparison of results reveals notable differences between the two methods. For concrete floor systems, the estimated number of HAP in Darwin increased from 3411 (percentage-based) to 11,473 (GIS-based), and in NSW, from 134,405 to 138,911. For timber floor systems, the HAP in Darwin increased from 2906 to 4043, while in NSW, it decreased significantly from 132,552 to 44,680 (Figure 7).

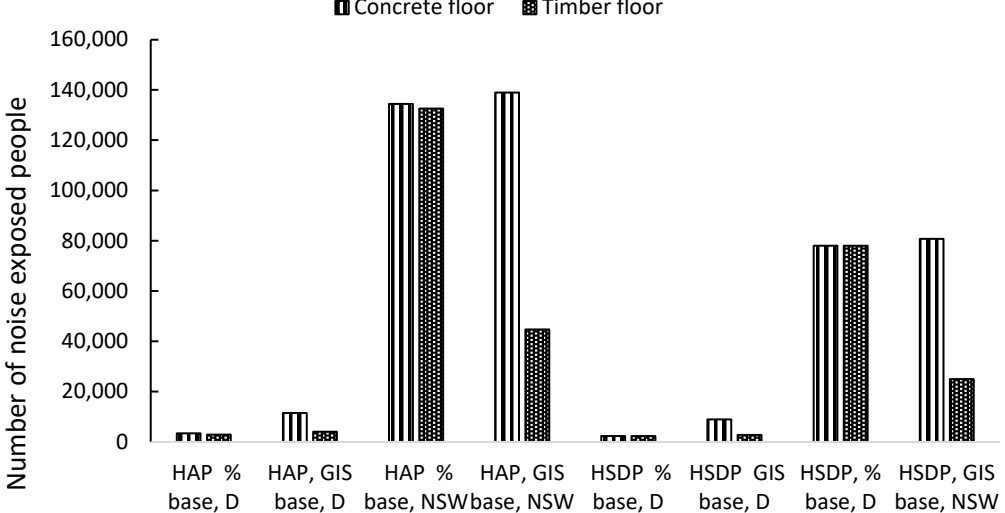

Comparative result of noise impact variation due to percentage base
and GIS base population calculation.

**Figure 7.** Noise-exposed population variation due to percentage- and GIS-based calculation.

A similar variation was observed in HSDP estimates. Under the percentage-based method, HSDP in Darwin and NSW were estimated at 2323 and 78,027, respectively. However, when using the GIS-based method for concrete floor construction, these figures increased to 8953 for Darwin and 80,747 for NSW. For timber floor systems, GIS-based HSDP estimates were 2771 in Darwin and 24,993 in NSW.

These differences are attributed to the increased accuracy of the GIS method, which accounts for actual spatial population distribution and excludes non-residential or un-populated zones. In contrast, the uniform percentage method may overestimate or un-derestimate population exposure, particularly in geographically and demographically heterogeneous areas.

While the percentage-based method remains valid for consistent comparisons across scenarios and is aligned with practices in similar LCA-based noise assessments, the GIS-based results add depth and spatial accuracy to the findings. Therefore, this dual-method approach strengthens the reliability of the study and highlights the potential for GIS integration in future construction noise assessments.

3.1.2. Circular Economy Scenarios: Reuse, Recycling, and Transport Distance Reduction

In this study, the authors also assessed the environmental impact of circular economy strategies by comparing two end-of-life scenarios: landfill and reuse or recycling. The percentage of recycling can vary up to 100% [42]. In this research, the authors chose 100% landfill disposal and 100% material reuse or recycling for timber and concrete floors. Reusing or recycling materials significantly reduces environmental burdens (Figure 8), making it a sustainable option aligned with circular economy principles. For reused concrete, no mineral extraction is required, and only energy is needed to grind demolished material for

reuse. A transport distance of 800 km and 50 km was assumed for virgin steel and timber material extraction, while a shorter 50 km haul was assumed for reused materials, reducing heavy vehicle traffic and thereby decreasing associated noise exposure.

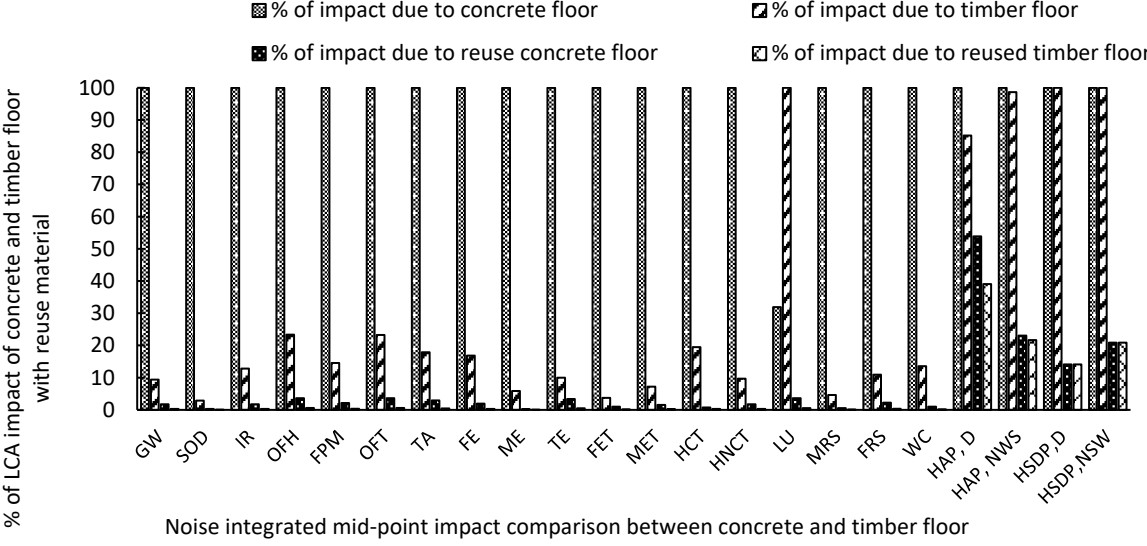

**Figure 8.** Comparative noise-integrated midpoint impacts of concrete and timber floor vs. reused concrete and timber floor. GW = global warming; SOD = stratospheric ozone depletion; IR = ionic radiation; OFH = ozone formation, human health; FPM = fine particulate matter formation; OFT = ozone formation, terrestrial ecosystems; TA = terrestrial acidification; FE = freshwater eutrophication; ME = marine eutrophication; TE = terrestrial ecotoxicity; FET = freshwater ecotoxicity; MET = marine ecotoxicity; HCT = human carcinogenic toxicity; HNCT = human non-carcinogenic toxicity; LU = land use; MRS = mineral resource scarcity; FRS = fossil resource scarcity; WC = water consumption; HAP, D = highly annoyed people in Darwin; HAP, NSW = highly annoyed people in NSW; HSDP, D = highly sleep-deprived people in Darwin; HSDP, NSW = highly sleep-deprived people in NSW; LCA = life cycle assessment.

All results in Figure 8 are presented using normalized midpoint values, where 100% represents the highest environmental impact value for each category across all scenarios. This comparative approach allows for easy visualization of relative performance. The normalization method follows standard LCA practice by converting raw impact data into dimensionless scores scaled against a reference value, as detailed in Table A7 (see Appendix A). This table provides the underlying characterization results and normalization factors applied to both environmental and noise-related midpoint indicators.

The life cycle assessment (LCA) results reveal that timber flooring significantly outperforms concrete in most environmental categories. New timber floors reduce emissions by 90.6% compared to new concrete for global warming potential, while reused timber achieves a remarkable 99.7% reduction. Substantial benefits are also seen in categories like stratospheric ozone depletion (97.1% lower in timber, 99.95% in reused timber), fine particulate matter formation (85.4% and 99.7% lower), and marine eutrophication (94.2% and 99.9% lower). Timber also cuts human non-carcinogenic toxicity by 90.3%, with reused timber providing an even stronger cut of 99.7%. Additionally, timber reduces fossil resource scarcity by 89.1% and water consumption by 86.4%. The only trade-off is land use, where timber shows a 213% increase over concrete due to forest resource demands (Figure 8).

The reused timber scenarios demonstrate remarkable advantages for noise impact, particularly in reducing community annoyance and sleep disturbances. In Darwin, the number of highly annoyed people (HAP) decreases from 3411 for new concrete to just 1332 in the reused timber scenario—representing a 60.9% reduction (see Appendix A:

Table A7). Similarly, in New South Wales (NSW), reused timber lowers HAP from 134,405 (new concrete) to 29,126 (reused concrete), marking a dramatic 78.3% decrease (Figure 6). Regarding sleep disturbance—a critical public health concern—the benefits of reused timber are equally compelling. In Darwin, the number of highly sleep-deprived people (HSDP) decreases from 2323 (new concrete) to just 328 (reused concrete), resulting in an 85.9% reduction (Figure 8). The same percentage drop is observed in NSW, where HSDP falls from 78,027 to 16,283.

Although the 100% recycling scenario is an idealized case and may not reflect current real-world conditions, it was used to determine the maximum possible environmental benefits through complete material recovery. To address practical limitations, a more realistic scenario was also assessed, incorporating 80% recycling and 20% landfill (see Appendix A: Table A8). This mixed end-of-life approach showed substantial impact reductions—about 80% for most environmental indicators. Regarding noise-related health impacts, the highly annoyed population (HAP) and highly sleep-disturbed population (HSDP) declined by 63–69% and 37–62%, respectively. These results offer a more balanced perspective on the environmental improvements achievable under realistic recycling conditions.

*3.2. Endpoint Impact Assessment*

The ReCiPe 2016 Endpoint (H) impact assessment method was employed to evaluate the long-term consequences of construction activities across three key areas of protection: human health, ecosystems, and resources. Compared to midpoint indicators, endpoint assessments are particularly valuable because they translate complex environmental emissions into tangible, decision-relevant outcomes—such as years of life lost or species affected. This enables more effective communication of environmental trade-offs, particularly when public health concerns like noise exposure are considered. Traditional endpoint LCAs capture human health damage largely through emissions-based pathways (e.g., air pollution, toxicity), but they rarely quantify direct health burdens from noise exposure. By calculating disability-adjusted life years (DALYs) from both material-related emissions and noise exposure, this study pioneers an integrated endpoint analysis that more fully represents construction's total impact on human health. While endpoint methods involve greater uncertainty due to value-based assumptions, they are essential for policy-oriented and holistic life cycle assessments. Accordingly, the hierarchist perspective was selected, representing a balanced and widely accepted scientific viewpoint [62].

Among the three endpoint categories, human health was prioritized in this study, given the direct and indirect health consequences of construction noise. Human health impacts were quantified using disability-adjusted life years (DALYs), a metric that reflects years of healthy life lost due to disease, disability, or premature death. Baseline DALY values—excluding noise—were obtained using the ReCiPe 2016 Endpoint (H) method in SimaPro. The results indicated 0.029 DALYs for concrete and 0.00392 DALYs for timber, based on material acquisition, production, construction, use, maintenance, and end-of-life phases. For noise impact assessment, derivation of the DALY value is described in Appendix A.3. After integrating noise impacts, the total human health burden in Darwin was 0.031149 DALYs for concrete (0.029 from material, 0.0011 from transport noise, and 0.000093 from equipment noise) and 0.0051 DALYs for timber (0.00392 from material, 0.0011 from transport noise, and 0.000092 from equipment noise (Table A9)). In contrast, the total impact in New South Wales (NSW) was significantly higher due to population exposure, with concrete floors resulting in 0.07757 DALYs (0.029 from material, 0.04885 from transport noise, and 0.000392 from equipment noise) and timber floors showing 0.05248 DALYs (0.00392 from material, 0.04885 from transport noise, and 0.00032 from equipment noise).

These findings reveal that the environmental burden of concrete construction is more than seven times greater than that of timber when noise is excluded (Figure 9). In Darwin, material production accounts for 93.1% of the total impact for concrete and 77% for timber. However, where transportation-related noise exposure is higher in NSW, material-related contributions drop to 37.45% for concrete and 7.5% for timber. These results highlight the significance of integrating noise into life cycle assessments, especially in densely populated regions. Equipment-related noise impacts remain comparatively minimal, contributing between 0.3% and 1.5% of total DALYs across both locations.

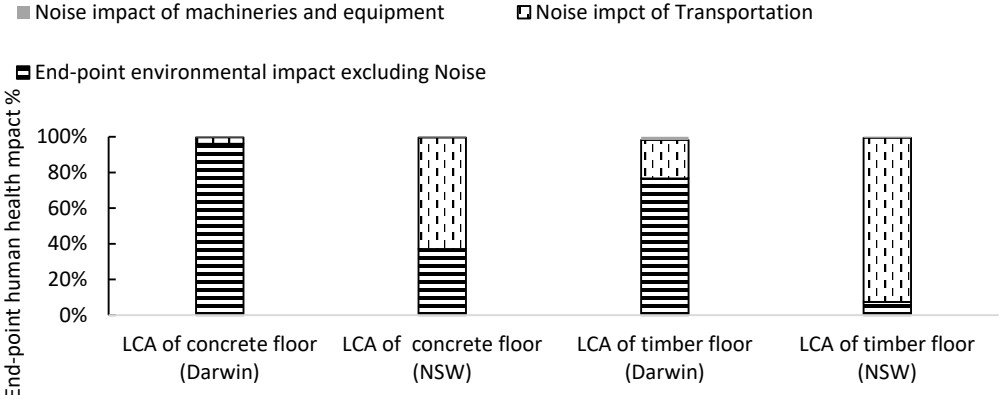

**Figure 9.** Noise-integrated endpoint impact variation between concrete and timber floor.

### 3.3. Sensitivity Analysis

This research concludes the significance of noise integration in LCA assessment. Noise reduction can reduce the environmental impact. Noise mitigation measures such as double glazing can reduce 10 dB [63,64]. As a result, the impact of concrete noise will be reduced by 2% and 31% in Darwin and NSW, respectively (Figure 10). The effect of timber noise will be reduced by 10% and 45% in Darwin and NSW, respectively. Irrespective of any material, noise mitigation has less effect in low-populated areas such as Darwin. On the contrary, noise mitigation measures significantly impact highly populated areas such as NSW.

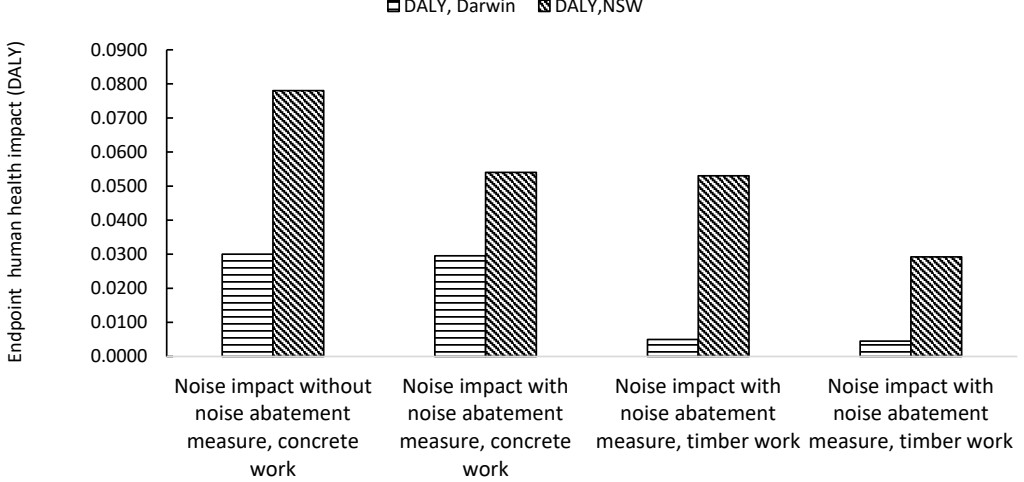

**Figure 10.** Impact of noise impact variation due to noise barrier.

*3.4. Results Validation*

3.4.1. Material Impact, Specifically Carbon Emission

The midpoint impact results of the LCA for concrete and timber in this study are entirely consistent with findings from recent comprehensive literature reviews. This research calculates the carbon emissions at 411 kg $CO_2$ eq./m$^2$ for concrete and 39 kg $CO_2$ eq./m$^2$ for timber. These results fall well within the range reported in similar studies, where the carbon footprint for concrete and timber structures typically varies between 90 and 800 kg $CO_2$ eq./m$^2$ [65]. Timber–concrete composite floor emits 38–74 kg $CO_2$ eq./m$^2$ and depends on the end-of-life stage [66]. For a concrete wall, carbon emission is three times higher (327.8 kg $CO_2$ eq./m$^2$) compared to a timber wall (117.2 kg $CO_2$ eq./m$^2$) [67]. However, for high-rise concrete structure buildings, carbon emissions can vary between 2430–2917 kg $CO_2$ eq./m$^2$ [68].

3.4.2. Traffic Noise Validation

Previous studies have assessed transportation-related noise impacts using a 500 km travel distance with a 40 ton truck, reporting human health endpoint impacts (DALYs) of 0.00065 for annoyance and 0.0066 for sleep deprivation [17]. In the present research, a longer distance of 1600 km with 16–32 ton lorries was considered, resulting in DALY values ranging from 0.00037 to 0.02435 for annoyance and from 0.00073 to 0.025 for sleep deprivation. The lower end of this range corresponds to Darwin, a low-population-density region, while the higher end reflects New South Wales (NSW), where population exposure is substantially greater. When adjusted for the same 1600 km distance, previous studies reported DALY values of 0.0002345 for annoyance and 0.024 for sleep deprivation, which aligns closely with the present findings. Additionally, another supporting study applied disability weights of 0.02 for annoyance and 0.07 for sleep deprivation, yielding noise impacts of 0.0017 and 0.0028 DALYs per km, respectively [58]. In the current research, disability weights of 0.0033 for annoyance and 0.0055 for sleep deprivation were applied, following WHO recommendations. By scaling the previous study's results based on the ratio of disability weights, the adjusted impacts are 0.00028 DALYs for annoyance and 0.00022 DALYs for sleep deprivation per km. When extended to the transportation distances analyzed here, the recalculated DALY values are 0.01 for annoyance and 0.036 for sleep deprivation. These consistent findings with international benchmarks validate the robustness of the current noise impact assessment and reinforce the reliability of the health burden estimates reported in this study.

3.4.3. Construction Nosie Validation

In this study, the disability weight for annoyance was initially set at 0.0033, leading to DALY values ranging from 0.000096 to 0.000382, based on an exposed population of between 1439 and 5117 people. In contrast, other researchers have used a higher disability weight, with DALY values for construction noise impacts reported between 0.8 and 27.977 per one million people (equivalent to 0.0008–0.027977 per person) [69]. According to their estimates, the DALY range should fall between 0.00012 and 0.14 for the exposed population considered in the present study. To align with this benchmark, the initial DALY results were adjusted by applying a correction factor of 6.06 (derived from 0.02/0.0033). After this adjustment, the revised DALY values range from 0.00059 to 0.00199. This updated range falls within the expected scale reported in the literature, validating the reliability and comparability of the present study's results.

## 4. Limitations

There are some limitations in this research. (1) Data unavailability of transportation, equipment, and machinery variation: Due to the absence of transportation data for Darwin and NSW, the authors of this article rely on overseas data. Depending on the type of road, the slope, and the type of vehicle, the amount of noise generated by transportation can vary. In addition, the noise levels may vary depending on the contemporary technology, equipment, and apparatus. (2) Health damage associated with noise emissions is generally non-linear, and threshold noise levels are often applied to evaluate their effects. Certain types of noise, such as blast noise commonly observed in mining areas, are not considered in the present noise impact assessment. This exclusion is due to the fact that such noise events are sporadic, highly localized, and do not represent continuous or typical construction-related noise emissions. As a result, their contribution to long-term population exposure and cumulative health damage is considered negligible within the scope of this study. (3) An accurate noise map is essential to calculate the noise-affected population. Preparing a noise map for transportation over the road network is extensive work. Only significant roads are considered most of the time. (4) An accurate factory or construction site noise map is also crucial. Building orientation, configuration, and materials significantly affect noise map preparation.

## 5. Conclusions

This study has introduced a methodological innovation by integrating both traffic and construction noise into the life cycle assessment (LCA) framework for concrete construction, offering a more holistic and realistic evaluation of environmental and health impacts. Traditional LCA studies have largely focused on carbon emissions, energy use, and material depletion, while neglecting noise pollution—a critical factor affecting both environmental quality and human health. By incorporating noise impacts from both stationary (occupational) and mobile (traffic) sources, this research fills an important gap in sustainable construction assessment.

The comparative analysis between concrete and timber flooring revealed significant differences in environmental performance. Conventional LCA results demonstrated that concrete floors exert a higher environmental burden across most impact categories, up to 7.4 times greater than timber, except in land use. When noise impacts were integrated, the results varied with population density. In low-density regions like Darwin, noise contributed modestly (7–33%) to the overall impacts, whereas in high-density areas such as NSW, noise impacts were much more substantial, contributing between 62% and 92%. This highlights the necessity of considering local context when assessing construction impacts, particularly in urban development. Moreover, equipment-related (static) noise impacts were found to be comparatively minimal, contributing only between 0.3% and 1.5% of total DALYs across both locations. This indicates that traffic noise is the dominant factor in noise-related health burdens, while construction equipment noise has a negligible effect by comparison.

The study also found that timber flooring slightly reduces the number of highly annoyed individuals compared to concrete, although levels of sleep disturbance remain similar across both materials. End-of-life scenarios showed significant potential for impact reduction: reusing materials such as concrete and timber led to a 67–99.78% decrease in midpoint environmental impacts, underscoring the importance of circular economy practices in construction.

Furthermore, a sensitivity analysis was conducted to evaluate the effectiveness of noise mitigation measures, such as double-glazed windows and noise barriers, in reducing noise-related health impacts. The results indicated 2–10% reductions in low-density areas

and 31–45% reductions in high-density regions, highlighting the substantial benefits of targeted interventions. This analysis confirms that incorporating mitigation strategies can significantly lower LCA-based impact scores, reinforcing the importance of integrating noise into environmental assessments. Additionally, the material and noise impact results were independently validated, further enhancing the credibility and robustness of the proposed methodological framework.

Overall, this research underscores the importance of integrating noise into LCA to better inform sustainable construction practices and policy decisions. By providing a comprehensive framework that considers both material and noise impacts across the entire life cycle, this study supports more responsible material selection and construction planning, particularly in noise-sensitive urban environments. Future research should aim to refine noise emission inventories, improve modeling approaches, and extend the framework to other construction types and geographical contexts, thereby advancing global efforts toward sustainable and health-conscious building practices.

**Author Contributions:** Conceptualization, A.R. and T.K.; methodology, R.S.; software, R.S.; validation, T.K. and A.R.; formal analysis, R.S.; investigation, R.S.; resources, R.S.; data curation, R.S.; writing—original draft preparation, R.S.; writing—review and editing, T.K. and R.S.; visualization, T.K. and R.S.; supervision, A.R. and T.K. All authors have read and agreed to the published version of the manuscript.

**Funding:** This work is supported through an Australian Government Research Training Program Scholarship.

**Institutional Review Board Statement:** Not applicable.

**Informed Consent Statement:** Not applicable.

**Data Availability Statement:** The original contributions presented in this study are included in the article. Further inquiries can be directed to the corresponding authors.

**Conflicts of Interest:** The authors declare that they have no known competing financial interests or personal relationships that could have appeared to influence the work reported in this review article.

## Nomenclature

| | |
|---|---|
| LCA | life cycle assessment |
| LCIA | life cycle impact assessment |
| DALYs | disability-adjusted life years |
| BIM | building information modeling |
| ISO | International Organization for Standardization |
| DW | disability weight |
| HAP | highly annoyed person |
| HSDP | highly sleep-disturbed person |
| vkm | vehicle-kilometer |

## Appendix A

*Appendix A.1. Estimation of the Noise Level of the Point/Stationary/Static Source*

Distinct types of noise can be generated from a point source (construction machine), such as steady and non-steady noise. Steady noise shows a low temporal variation, such as noise generated from air compressors and asphalt finishers. Non-steady noises are fluctuating noises generated from concrete mixers and concrete plants, such as impulse noise or discrete impulse noise (hammer machine and pistol), intermittent noise (demolition machine), and quasi-steady noise (breaker and jackhammer) [46]. Noise depends on the mechanical parameter of the equipment, such as the hardness of the drill bit, which is

inversely proportional to the noise level [47]. Foundation work with concrete work is responsible for more than 25% of the noise. Foundation work with piling is responsible for 42.2% of noise generation [46]. The noise level of a stationary source depends on the equipment's noise emission level, the distance from the source, and the attenuation. Noise can be generated inside the industry or outside the environment, such as at construction sites. Noise impacts depend on the receiver's location. The receiver can also be present inside or outside the factory.

Figures A1 and A2 indicate the industrial noise propagation (indoor and outdoor systems). In industry, several equipment/machines are responsible for noise generation. If any person drives the equipment or machine, that person will be receiver 1. If personnel work or stay near the equipment or machine, those persons will be receiver 2. If any noise propagates through the boundary wall, the receiver will be assumed as receiver 3. If any machine or equipment is used in the open construction place, noise propagation is like transportation noise (Figure A2). There are also three diverse types of receivers. Receiver 1 will take the structural noise, receiver 2 (outdoor) will be exposed to airborne noise, and receiver 3 (indoor) will also be exposed to airborne noise. Although receiver 3 will be less exposed than another receiver, depending on noise level, it can adversely impact sleep deprivation. Thus, general equipment noise can be described as

$$
\begin{aligned}
Total\ static\ noise = \ &Structural\ noise + airborne\ noise + indoor\ noise\ of\ industry \\
&+ indoor\ noise\ of\ residential\ house
\end{aligned}
\tag{A1}
$$

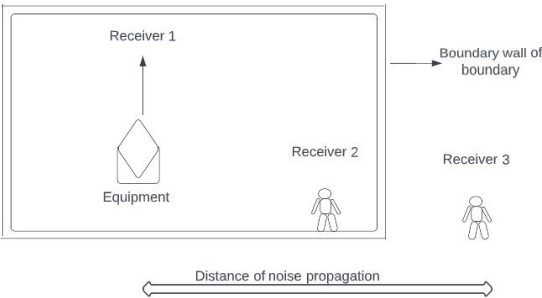

**Figure A1.** Equipment noise propagation around the industry.

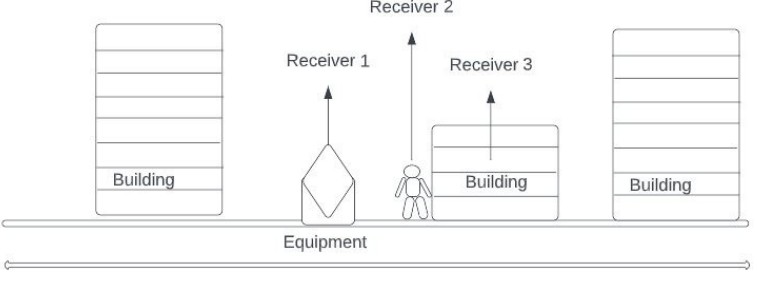

**Figure A2.** Equipment noise propagation around the construction site.

There are two types of noise, ground-borne and structure-borne noise, generated during construction work (Figure A3) [48]. The NSW Interim Construction Noise Guideline (EPA, 2009) provides residential noise management levels for regenerated noise for the evening (40 dB) and night-time (35 dB) periods. Where the vibration source interfaces directly with the structure (for example, a piece of mechanical plant or a hammer drill), the resulting re-radiated noise is called structure-borne noise [48]. Regenerated noise levels are

related to vibration velocity levels of the radiating surfaces. They are typically estimated using the following equation:

$$L_p = L_v - k \tag{A2}$$

where $L_p$ is the sound pressure level (dB re 20 μPa), $L_v$ is the spatially averaged vibration velocity level (dB re $1 \times 10^{-6}$ mm/s), and k is a constant for the receiving space, between 27 and 32 dB, which can be further expressed as follows:

$$LA_{eq}1 \, y \, (structural \, noise) = L_v - k \tag{A3}$$

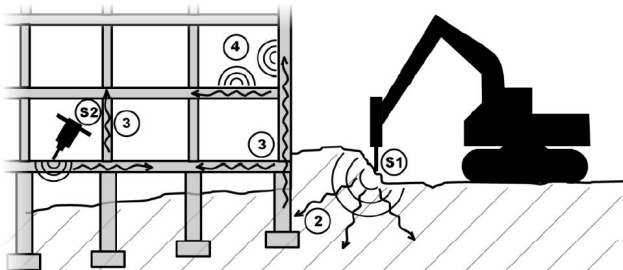

**Figure A3.** Ground-borne and structure-borne noise source. S1 and S2 is the structural noise source. Number 2, 3, and 4 are noise exposed area. The arrow sign indicates the noise propagation path.

The sound reduction also depends on the mass of the wall, floor, roof material, and room volume [11]. Structural noise depends on the mobility of the system ($Y_i$), frequency ($f_c$), longitudinal velocity ($C_l$), Young's modulus I, and dynamic stiffness ($s'$) [49].

Airborne noise propagation depends on distance, duration of equipment use, and screening [50]. It can be calculated as follows:

$$LA_{eq,2} \, y(airborne \, noise) = E.L + 10log(U.F) - 20log\frac{D}{50} - A \tag{A4}$$

where $LA_{eq,2}$ y(*airborne noise*) = the noise pressure level at a peak-hour period, E.L = the noise pressure level of the source at a reference distance of 50 feet, U.F = a usage factor that accounts for the fraction of time that the noise source is in use over the specified period in full power, D = the distance from the receiver to the noise source in feet, and A = the noise attenuation due to different types of screening (i.e., buildings, structures, barriers, etc.).

According to ISO 9613, there are distinct factors related to attenuation, such as geometry, atmosphere (weather, temperature, climate, and wind), diffraction, and reflection [51–53].

If a stationary noise source is enclosed inside the room boundary, the sound power level of the structure can be expressed as follows [54]:

$$LA_{eq,3} \, y(indoor \, noise \, in \, the \, industry) = LA - R + 10\frac{S}{A} \tag{A5}$$

where $LA_{eq,3}$ y is the A-weighted noise of the noise source, R is the in situ noise reduction index structure (decibel), S is the surface area of the structure ($m^2$), and A is the equivalent noise absorption surface of the structure ($m^2$). A can be derived from the following equation:

$$A = \sum_{i=1}^{n} \propto i \, . \, Si \tag{A6}$$

where i is the number of noise sources, such as machines, and $\alpha$ is the sound absorption coefficient.

The resultant sound power from the roof, wall, and other parts is as follows:

$$LA_{eq,3}y(indoor \, noise \, in \, industry) = 10log_{10}(0.1Leq(roof) + 0.1Leq(wall) + 0.1Leq(others)) \tag{A7}$$

In this main research work, Equation (1) has been used to calculate indoor noise in the industrial area.

$$LA_{eq,4} \, y(indoor \, noise \, of \, residential \, house) = E.L + 10log(U.F) - 20log\frac{D}{50} - A(indoor) \quad (A8)$$

Equation (A8) is the same as Equation (A4). Here, attenuation (A) depends on the noise pressure level at the source and the location of the receiver. A can be express as follows:

$$A = 1.163\frac{V}{T} \quad (A9)$$

where V is the volume of the sound chamber, m$^3$; and T is the measured reverberation time, s. Thus, yearly equivalent equipment noise can be expressed as follows:

$$LA_{eq} \, y \, (static) = 10log\{LA_{eq,1} \, y(structural \, noise) + LA_{eq,2} \, y(airborne \, noise) + LA_{eq,3} \, y(indoor \, noise \, in \, industry) \\ + LA_{eq,4} \, y(indoor \, noise \, of \, residential \, house)\} \quad (A10)$$

*Appendix A.2. Noise Level Calculation*

In Appendix A.2, we will first describe static noise calculation and then dynamic/mobile noise level calculation. Static noise is generated by machines and equipment. It consists of three components: structural noise, airborne noise, and indoor noise calculation.

Appendix A.2.1. Noise Level in Construction Area

It is assumed that concrete and timber construction will be held in Gray, Darwin and Paddington, NSW. The population density (per km$^2$) is 3310 and 12,134 people, respectively. The noise level derivations are given below:

Noise Level for Concrete Work

A concrete mixer, compressor, compression, and compactor are used for concrete pouring. The average decibel level produced by these machines is 94.3 dB (Table A4). As per Equation (A3), the structural noise level is 67.3. This calculation assumes that the machine has been operating for four hours and that the environmental attenuation is 10 decibels. As per Equation (A4), the noise level at 150 feet for receiver 2 is 84.3 decibels. Assuming additional building wall attenuation of 10 dB, the noise level for the adjacent resident (receiver 3) will be 64.3 dB (Figure 4).

Noise Level for Timber Work

For timber work, an excavator, jackhammer, timber lifting crane, and chainsaw have been used, generating 87.5 dB (Table A4). Receivers 1, 2, and 3 are exposed to 60.5, 77.5, and 57.5 dB. These results indicate that timber work is less significant than concrete construction.

Appendix A.2.2. Estimation of the Noise Level of the Mobile Source

Motor and exhaust systems are the main factors in transportation noise generation [56]. Noise levels vary depending on the site (residential, industrial, commercial, rural) and time (day, evening, night).

Figures A4 and A5 illustrate the noise propagation from transportation. They indicate that if a transport system (truck) generates noise, there can be three diverse types of noise receivers: receiver 1 (driver or passenger of the truck), receiver 2 (pedestrian), and receiver 3 (people living in the house).

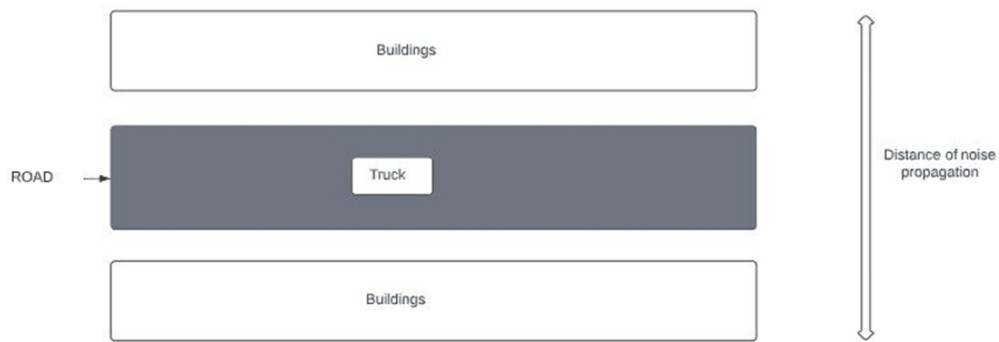

**Figure A4.** Noise propagation from the side of the road. The ash color is road which is also indicated with arrow.

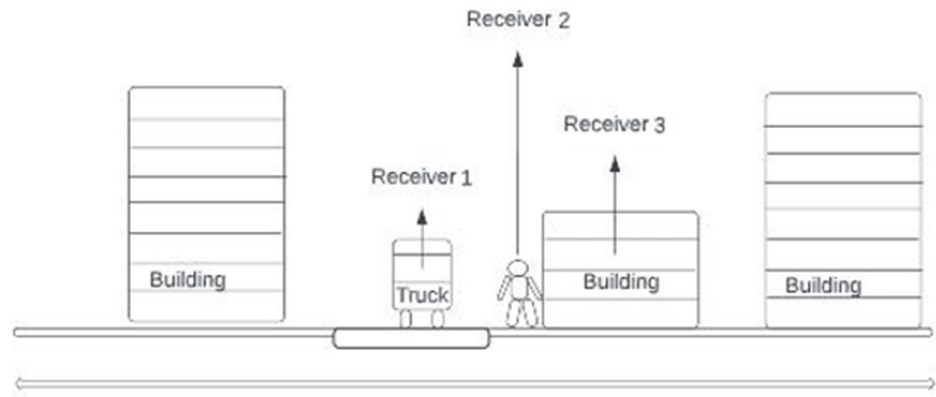

**Figure A5.** Transportation noise propagation.

As per Figures A4 and A5, there are three diverse types of noise receivers from transportation noise. Summation of transportation noise can be derived as follows:

$$Total\ transportation\ noise = Structural\ noise + airborne\ noise + indoor\ noise\ of\ residential\ house \qquad (A11)$$

Structural noise ($LA_{eq,\ 1}$ m) can be derived from Equation (A3).

As a mobile source, trucks and concrete mixture trucks are used for concrete work. To calculate airborne mobile noise ($LA_{eq,\ 2}$ m), the following equation has been used:

$$LA_{eq,2}m = 10log\left(10^{0.1\ x\ LE1} + 10^{0.1XLE2}\right) \qquad (A12)$$

$$LE_1 = E_1 + \log(N_1)$$

where $E_1 = max[\{12.8 + 19.5 \times \log(V_1)\}], \{45 + 0.8 \times (0.5i - 1.5)\}$ for car,

$$LE_2 = E_2 + \log(N_2)$$

where $E_2 = max[\{34 + 13.3 \times \log(V_2)\}], \{56 + 0.6 \times (0.5i - 1.5)\}$ for truck.

$N_1$ and $N_2$ are the average number of light vehicles (cars, vans, and light motorcycles) and heavy vehicles (trucks, buses, and tractors) per hour. $V_1$ and $V_2$ is the velocity. The road surface gradient is expressed as i.

Indoor noise of residential house ($LA_{eq,\ 3}$ m) can be derived from Equation (A8).

Total transportation noise will be as follows:

$$LA_{eq}\ m\ (mobile) = 10log\{LA_{eq,1}\ m(structural\ noise) + LA_{eq,2}\ m(airborne\ noise) \\ + LA_{eq,3}\ m(indoor\ noise\ of\ residential\ house)\} \qquad (A13)$$

The total $LA_{eq}$ (total) noise levels from construction noise sources can be combined using the following equation.

$$LA_{,total} = 10log(LA_{eq\ y\ (static)} + LA_{eq\ m(mobile)}) \qquad (A14)$$

where $LA_{eq,y\ (static)}$ = predicted yearly average noise level from construction equipment (airborne, structural, and vibration noise) and $LA_{eq,m\ (mobile)}$ = predicted yearly average noise level from a mobile source.

Traffic Data Input

The average number of vehicles per vehicle type, average speed, road type, and road properties (type of surface, gradient) are required to compute the traffic noise level ($LA_{eq}$ m). According to Figures A4 and A5, there are three distinct categories of transpositional noise receivers. These recipients include the driver and passengers, pedestrians, and nearby building residents. Transportation can generate two distinct noise categories, including structural and airborne noise. The driver and passengers are affected by structural noise caused by engine-generated vehicle vibration. Both pedestrians and building occupants are impacted by airborne noise. Therefore, calculating the noise attenuation between the road and the building facade is necessary to measure the noise level of the roadside area. In addition, other information such as building type, height, orientation, and substance is also required to calculate the noise level within a building interior.

Here, the authors have calculated using research data. It is presumed that one heavy vehicle and one light vehicle are used for transportation. The heavy vehicles included a truck for mineral delivery from mining to the industry, delivering concrete to the construction site, material delivery during the maintenance period, and demolished content delivery from the site to the landfill area. A small vehicle transports construction workers during every phase, such as mining, industry, construction and maintenance, and end-of-life phases. The distance and traveling time are listed in Table A5.

The structural noise level of trucks and other modes of conveyance is 64 decibels (Lv) (assumed). Here, the vehicle and the automobile generate 65 dB (assumed), and the constant k value is 27 dB. According to the following Equation (A3), the level of structural noise for receiver 1 is 37 dB.

Here, Darwin and Sydney are represented as rural and urban areas, respectively. Average truck and car speeds in the Darwin region are 80 km/h and 50 km/h, respectively, while in the NSW region, those speeds are 75 km/h and 50 km/h, respectively. The average number of trucks and cars per hour in the NSW region is 153 and 1626, while in the Darwin region, it is nine and 53 (similar data from Switzerland) [9]. To evaluate noise impact, we added one truck and one car to this baseline Swiss traffic flow, ensuring that the variation stays within small limits. According to ISO 1996-1:2016 [70] and WHO Environmental Noise Guidelines (2018) [69], changes within ±3 dB are considered small and suitable for linear noise prediction methods. Our calculated noise increase remains within this threshold, validating our approach. The distribution of traffic can alter during various times of the day, such as the day, evening, and night. Here, most of the concrete construction work is performed during the day. Therefore, temporal variation is not regarded in this assessment of noise. According to Equation (A12), the mean noise level for the pedestrian (receiver 2) in Darwin and NSW is 69.87 and 82.87 decibels, respectively. Noise level measurement for the indoor place (for receiver 3) can be calculated as per Equation (A8).

*Appendix A.3. Computation of Characterization Factors (CFs)*

Dose–response relationships link sleep disturbance to $L_{night}$, and noise annoyance to noise level, as shown in Equations (5) and (6) (main paper). These two health impairments are related to the night and day periods, respectively. Although there are three time-segments to calculate the noise impact, it is impossible to differentiate between evening and day. Thus, the CF is defined as 10 p.m. to 6 a.m. for the night period and 6 a.m. to 10 p.m. for the day/evening period. As the marginal approach is considered for traffic flow, it needs an elementary flow. Here, vkm is taken as elementary flow, which is obtained by summing up, over the three periods, i {day, evening, night} of the increase in vkm on an hourly scale and Δ traffic multiplied by the number of hours in the corresponding period $H_i$ and the number of days in one year. For example, the marginal increase in traffic Δvkm over one year is as follows:

$$\Delta vkm_{year, \ whole \ day} = \sum_i \Delta traffic_i H_i \tag{A15}$$

The resulting CF for annoyance is obtained using Equation (A15). EF is the elementary flow and refers to an increase in vkm. The number of HAP is calculated from the exposed population and the dose–response relationship for annoyance presented in Equation (A16).

$$CF_{HAP} = \frac{Number \ of \ HAP}{EF_{Year, \ whole \ day}} \tag{A16}$$

The same approach is applied to sleep disturbance. It is noted that $CF_{HSDP}$ is valid only at night, as expressed in Equation (A16). The number of HSPD is derived from the exposed population and the dose–response relationship for sleep disturbance presented in Equation (A17).

$$CF_{HSDP} = \frac{Number \ of \ HDSP}{EF_{Year, \ night}} \tag{A17}$$

The resulting endpoint CF in DALYs is calculated per Equations (A18) and (A19), where the midpoint CF is multiplied by the corresponding disability weight DW.

$$CF_{day, \ evening} = CF_{HAP} \times DW_{HAP} \tag{A18}$$

$$CF_{night} = CF_{HAP} \times DW_{HAP} + CF_{HSDP} \times DW_{HSDP} \tag{A19}$$

*Appendix A.4. List of the Tables*

**Table A1.** Initial bill of quantities of concrete floor.

| Type of Work | Name of Material | Quantity of Material | Service Time |
|---|---|---|---|
| Trenching | Soil digging | 40 m$^3$ | |
| | reinforcement | 1100 kg | 100 years |
| | 6 mm laminated floor panel | 1.044 m$^3$ | 20 years |
| Wooden floor maintenance | Sanding, vacuuming | | 10 years |
| | Oil-based polyurethane | 20 L | 10 years |

**Table A2.** Initial bill of quantities of timber floor.

| Type of Work | Name of Material | Quantity of Material | Service Time |
|---|---|---|---|
| Wooden floor | Structural timber of floor foundation | 3.4 m$^3$ | 100 years |
| | Nail for timber foundation | 6.9 kg | 100 years |
| | 20 mm wooden floor panel | 3.3 m$^3$ | 50 years |
| | Aluminum nail | 10 kg | 50 years |
| Wooden floor maintenance | Sanding, vacuuming | | 10 years |
| | Oil-based polyurethane | 20 L | 10 years |

**Table A3.** Activities and information required for final bill of quantities.

| Product Stage | Material Information |
|---|---|
| Packaging material: | Concrete information:<br>For 110,000 kg of reinforced concrete, 15,400 kg of cement, 30,800 kg of sand, 61,600 kg of crushed stone, 1100 kg of steel, and 6600 kg of water were used.<br>A 18.64 kw machine will operate for 8 h for concrete production.<br>A 16–32 metric ton lorry, EURO4 | Cut-off, U will travel 800 km from the mining place to the industry and 50 km to the construction site.<br>For 60,000 kg earth excavations, 18.64 kw machines will operate for 8 h.<br>Timber information:<br>For 3.4 m$^3$ of timber footing, glue-laminated timber is used, and a 6.9 kg aluminum nail is used for fixing. Cross-laminated timber is used for 3.3 m$^3$ of timber floor, and a 10 kg aluminum nail is used for fixing.<br>A 16–32 metric ton lorry, EURO4 | Cut-off, U will travel 50 km distance.<br>For 30,000 kg earth excavations, an 18.64 kw machine will operate for 4 h. |
| During the construction stage: | One kWh of electricity vibrates the concrete. Seventy kg of polyurethane rigid foam acts as a vapor barrier underneath the concrete. A 2800 kg vinyl floor covers the floor.<br>Hammer guns and hand saw machines are used for timber work. |
| Maintenance and repair stage: | An amount of 200 kg of anionic resin is used to maintain the floor covering, and 100 kg of wood preservative is used to protect the wooden floor. |
| Deconstruction and disposal stage: | Landfill scenario:<br>The concrete demolishing hammer is used for 8 h for concrete work, and leftover concrete will go to a landfill.<br>The landfill location is 100 km away from the construction site.<br>Reinforcement will be gone in the landfill, too.<br>The timber will be demolished by an 18.64 kW machine for 4 h and transferred to 100 km for landfill.<br>Reuse scenario:<br>The concrete is demolished with the machine for 16 h and will be reused. The reuse mechanism factory is 100 km away from the construction site.<br>Reinforcement will be reused, too.<br>The timber will be demolished by an 18.64 kW machine for 8 h and transferred to 100 km for landfill.<br>Nail scrap will be separated and sent to the factory.<br>Steel: Steel, low-alloyed {GLO} | market for | Cut-off, U<br>Transport, freight, lorry 16-32 metric ton, euro4 {RoW} |<br>Brick: Clay brick {GLO} | market for | Cut-off, U<br>Transport, freight, lorry 16–32 metric ton, euro4 {RoW} |<br>Insulation: Glass wool mat {GLO} | market for | Cut-off, U<br>Transport, freight, lorry 3.5–7.5 metric ton, euro6 {RER} |<br>Wood-based insulation: cellulose fiber {RoW} | market for cellulose fiber | Cut-off, U<br>Transport, freight, lorry 3.5–7.5 metric ton, euro5 {RoW} |<br>Paint: Alkyd paint, white, without solvent, in 60% solution state {RER} | market for alkyd paint, white, without solvent, in 60% solution state | APOS, S<br>Plasterboard: Gypsum plasterboard {GLO} | market for | Cut-off, U<br>Door: Door, outer, wood-glass {GLO} | market for | Cut-off, S<br>Window: Transport, freight, lorry 16–32 metric ton, EURO6 | Cut-off, U |

**Table A4.** List of noise sources of equipment.

| Name of Point Source | Decibel |
|---|---|
| Concrete work-related equipment | |
| Rock drill | 97 |
| Steel reinforcement forming for concrete | 90 |
| Aluminum forming and processing | 80 |
| Vibrating roller | 106 |
| Concrete mixer | 86 |
| Jackhammer | 87 |
| Construction lift | 93 |
| Pump | 100 |

**Table A4.** *Cont.*

| Name of Point Source | Decibel |
|---|---|
| Crawler excavators 0.9–9 tons | 97 |
| Crawler excavators 12–40 tons | 103 |
| Crawler piling rig | 110 |
| Skid-steer loaders | 101 |
| Excavator with a demolition hammer | 114 |
| Excavator | 76 |
| Timber work-related equipment | |
| Timber harvester | 75 |
| Forwarder | 82 |
| Self-loading tractor | 91 |
| Grapple skidder | 78 |
| Forest loader | 82 |
| Chainsaw | 100 |
| Timber lifting crane | 87 |
| Jackhammer | 87 |
| Excavator | 76 |

**Table A5.** Traveling distance and time of heavy vehicles (truck) for steel; traveling distance and time of light vehicles (passenger car).

| Type of Vehicle | Traveled from | Travel to | Total Distance | Traffic Velocity | Traveling Time |
|---|---|---|---|---|---|
| Heavy vehicle (truck) | Mining | Industry | (800 + 800) = 1600 km | 80 km/h | 20 h |
| | Industry | Construction site | (25 + 25) = 50 km | 50 km/h | 1 h |
| | Maintenance material factory | Construction site | (25 + 25) = 50 km | 50 km/h | 1 h |
| | Construction site | Landfill | (50 + 50) = 100 km | 50 km/h | 2 h |
| Passenger car | Mining | 5 | (25 + 25) = 50 km | 50 km/h | 5 h |
| | Industry | 5 | (25 + 25) = 50 km | 50 km/h | 5 h |
| | Construction site | 5 | (25 + 25) = 50 km | 50 km/h | 5 h |
| | Maintenance time | 2 | (25 + 25) = 50 km | 50 km/h | 2 h |
| | Demolition time | 2 | (25 + 25) = 50 km | 50 km/h | 2 h |
| | EOL factory | 5 | (25 + 25) = 50 km | 50 km/h | 5 h |

**Table A6.** Midpoint impact of concrete and timber.

| | Impact Category | Unit | LCA of Concrete with Cradle-to-Grave Life Cycle | LCA of Timber Floor with Cradle-to-Grave Life Cycle | % of Impact Due to Concrete Flooring | % of Impact Due to Timber Flooring |
|---|---|---|---|---|---|---|
| Global warming | GW | kg $CO_2$ eq | 71,584.59 | 6717.785 | 100 | 9.384401 |
| Stratospheric ozone depletion | SOD | kg $CFC_{11}$ eq | 0.129747 | 0.00374 | 100 | 2.8825329 |
| Ionic radiation | IR | kg $Co_{-60}$ eq | 256.4087 | 32.88057 | 100 | 12.8235 |
| Ozone formation, human health | OFH | kg $NO_x$ eq | 238.5943 | 55.68097 | 100 | 23.337091 |
| Fine particulate matter formation | FPM | kg PM 2.5 eq | 34.9953 | 5.096214 | 100 | 14.562567 |
| Ozone formation, terrestrial ecosystems | OFT | kg $NO_x$ eq | 245.0896 | 56.95245 | 100 | 23.2374 |
| Terrestrial acidification | TA | Kg $SO_2$ eq | 183.2309 | 32.7833 | 100 | 17.891797 |
| Freshwater eutrophication | FE | kg P eq | 1.565374 | 0.263861 | 100 | 16.8561 |
| Marine eutrophication | ME | kg N eq | 1.735597 | 0.101414 | 100 | 5.8431767 |
| Terrestrial ecotoxicity | TE | kg 1,4-DCB | 217,716.9 | 21,668.13 | 100 | 9.9524336 |
| Freshwater ecotoxicity | FET | kg 1,4-DCB | 305.2816 | 11.31409 | 100 | 3.7061159 |
| Marine ecotoxicity | MET | kg 1,4-DCB | 177.7454 | 12.82773 | 100 | 7.2169125 |
| Human carcinogenic toxicity | HCT | kg 1,4-DCB | 63.21805 | 12.3086 | 100 | 19.470072 |
| Human non-carcinogenic toxicity | HNCT | kg 1,4-DCB | 1368.211 | 132.7463 | 100 | 9.7021804 |
| Land use | LU | $m^2$a crop eq | 3343.709 | 10,476.46 | 31.9164 | 100 |
| Mineral resource scarcity | MRS | kg Cu eq | 448.0035 | 20.583 | 100 | 4.5943837 |
| Fossil resource scarcity | FRS | kg oil eq | 18,034.2 | 1968.984 | 100 | 10.918056 |
| Water consumption | WC | $m^3$ | 422.643 | 57.28129 | 100 | 13.553115 |
| Noise, highly annoyed people, Darwin | HAP, D | HAP | 3411 | 2906 | 100 | 85.184221 |
| Noise, highly annoyed people, NSW | HAP, NWS | HAP | 134,405 | 132,552 | 100 | 98.621674 |
| Noise, highly sleep-deprived people, Darwin | HSDP, D | HSDP | 2323 | 2323 | 100 | 100 |
| Noise, highly sleep-deprived people, NSW | HSDP, NSW | HSDP | 78,027 | 78,027 | 100 | 100 |

**Table A7.** Midpoint impact of recycled concrete and timber.

| Impact Category | Unit | LCA of Concrete with Cradle-to-Grave Life Cycle | LCA of Timber Floor with Cradle-to-Grave Life Cycle | LCA of Reused Concrete with Cradle-to-Grave Life Cycle | LCA of Reused Timber Floor with Cradle-to-Grave Life Cycle |
|---|---|---|---|---|---|
| Global warming | kg $CO_2$ eq | 71,584.59 | 6717.785 | 1225.318 | 199.1238 |
| Stratospheric ozone depletion | kg $CFC_{11}$ eq | 0.129747 | 0.00374 | 0.000376 | $6.36 \times 10^{-5}$ |
| Ionic radiation | kg $Co_{-60}$ eq | 256.4087 | 32.88057 | 4.452854 | 0.725207 |
| Ozone formation, human health | kg $NO_x$ eq | 238.5943 | 55.68097 | 8.670905 | 1.401802 |
| Fine particulate matter formation | kg PM 2.5 eq | 34.9953 | 5.096214 | 0.739111 | 0.116319 |
| Ozone formation, terrestrial ecosystems | kg $NO_x$ eq | 245.0896 | 56.95245 | 8.861374 | 1.431817 |
| Terrestrial acidification | Kg $SO_2$ eq | 183.2309 | 32.7833 | 5.306071 | 0.837611 |
| Freshwater eutrophication | kg P eq | 1.565374 | 0.263861 | 0.030383 | 0.004391 |
| Marine eutrophication | kg N eq | 1.735597 | 0.101414 | 0.003791 | 0.00058 |
| Terrestrial ecotoxicity | kg 1,4-DCB | 217,716.9 | 21,668.13 | 7284.772 | 1044.376 |
| Freshwater ecotoxicity | kg 1,4-DCB | 305.2816 | 11.31409 | 2.991508 | 0.437 |
| Marine ecotoxicity | kg 1,4-DCB | 177.7454 | 12.82773 | 2.742173 | 0.399516 |
| Human carcinogenic toxicity | kg 1,4-DCB | 63.21805 | 12.3086 | 0.450204 | 0.175154 |
| Human non-carcinogenic toxicity | kg 1,4-DCB | 1368.211 | 132.7463 | 23.47352 | 3.873507 |
| Land use | $m^2$a crop eq | 3343.709 | 10,476.46 | 120.8077 | 17.19549 |
| Mineral resource scarcity | kg Cu eq | 448.0035 | 20.583 | 2.402292 | 0.359573 |
| Fossil resource scarcity | kg oil eq | 18,034.2 | 1968.984 | 399.8294 | 64.52096 |
| Water consumption | $m^3$ | 422.643 | 57.28129 | 4.085829 | 0.598325 |
| Noise, highly annoyed people, Darwin | HAP, D | 3411 | 2906 | 1838 | 1332 |
| Noise, highly annoyed people, NSW | HAP, NWS | 134,405 | 132,552 | 30,979 | 29,126 |
| Noise, highly sleep-deprived people, Darwin | HSDP, D | 2323 | 2323 | 328 | 328 |
| Noise, highly sleep-deprived people, NSW | HSDP, NSW | 78,027 | 78,027 | 16,283 | 16,283 |

**Table A8.** Impact of mix ratio of recycle and landfill scenario.

| Midpoint LCA Impact | Mix Ratio of Concrete Impact | Mix Ratio of Timber Impact |
|---|---|---|
| kg $CO_2$ eq | 15,297.17 | 1502.86 |
| kg $CFC_{11}$ eq | 0.03 | 0.00 |
| kg $Co_{-60}$ eq | 54.84 | 7.16 |
| kg $NO_x$ eq | 54.66 | 12.26 |
| kg PM 2.5 eq | 7.59 | 1.11 |
| kg $NO_x$ eq | 56.11 | 12.54 |
| Kg $SO_2$ eq | 40.89 | 7.23 |
| kg P eq | 0.34 | 0.06 |
| kg N eq | 0.35 | 0.02 |
| kg 1,4-DCB | 49,371.20 | 5169.13 |
| kg 1,4-DCB | 63.45 | 2.61 |
| kg 1,4-DCB | 37.74 | 2.89 |
| kg 1,4-DCB | 13.00 | 2.60 |
| kg 1,4-DCB | 292.42 | 29.65 |
| $m^2$a crop eq | 765.39 | 2109.05 |
| kg Cu eq | 91.52 | 4.40 |
| kg oil eq | 3926.70 | 445.41 |
| $m^3$ | 87.80 | 11.93 |
| HAP, D | 2152 | 1647 |
| HAP, NWS | 51,664 | 49,811 |
| HSDP, D | 727 | 727 |
| HSDP, NSW | 28,632 | 28,632 |

**Table A9.** Endpoint impact percentage of concrete and timber floor.

| | LCA of Concrete Floor (Darwin) | LCA of Concrete Floor (NSW) | LCA of Timber Floor (Darwin) | LCA of Timber Floor (NSW) |
|---|---|---|---|---|
| Endpoint environmental impact excluding noise | 96.0 | 37.1 | 76.7 | 7.4 |
| Noise impact of transportation | 3.6 | 62.5 | 21.5 | 92.0 |
| Noise impact of machineries and equipment | 0.3 | 0.4 | 1.8 | 0.6 |

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
