# Peer review of "Integrating Noise Pollution into Life Cycle Assessment: A Comparative Framework for Concrete and Timber Floor Construction"

_sustainability, doi:10.3390/su17146514_

Round 1
Reviewer 1 Report (Previous Reviewer 1)
Comments and Suggestions for Authors
The revised manuscript has been improved significantly after the revision. The response for the reviewers's comments has been also well done. It may be accepted now.
Author Response
Reviewer's comment :The revised manuscript has been improved significantly after the revision. The response for the reviewers's comments has been also well done. It may be accepted now.
Answer: Many thanks for your comments and guidelines.
Reviewer 2 Report (Previous Reviewer 3)
Comments and Suggestions for Authors
The current version of the paper is well written and ready for publication.
Author Response
Comment: The current version of the paper is well written and ready for publication.
Response: Many thanks for your valuable feedback
Reviewer 3 Report (Previous Reviewer 2)
Comments and Suggestions for Authors
I have received a new submission of this article for review. I have already reviewed the article, which raised a number of fundamental issues affecting the basis of the research, such as the illogical hypothesis of replacing 100% of the aggregates, concrete transport distances of more than 1,000 km, and the insufficient assessment of people affected by traffic. Now it is said that a new study has been carried out, but its basis is not indicated, etc. All of this significantly affects the initial hypotheses of the research and, therefore, the results and conclusions. I therefore regret to propose its rejection.
Author Response
Comments: I have received a new submission of this article for review. I have already reviewed the article, which raised a number of fundamental issues affecting the basis of the research, such as the illogical hypothesis of replacing 100% of the aggregates, concrete transport distances of more than 1,000 km, and the insufficient assessment of people affected by traffic. Now it is said that a new study has been carried out, but its basis is not indicated, etc. All of this significantly affects the initial hypotheses of the research and, therefore, the results and conclusions. I therefore regret to propose its rejection.
Response :
Many thanks for your valuable feedback: There are 3 major point to discuss,
1) Recycle percentage of building material.
2) Material transport distance
3) Noise exposed people calculation.
Here is my answer:
1) Recycle percentage (section 3.1.2, page 16)
“Although the 100% recycling scenario is an idealised case and may not reflect current real-world conditions, it was used to establish the maximum potential environmental benefits achievable through complete material recovery. To address the practical limitations of 100% recycling, a more realistic scenario was also evaluated, incorporating 80% recycling and 20% landfill. This mixed end-of-life strategy demonstrated significant impact reductions—approximately 80% for most environmental indicators. In terms of noise-related health impacts, the Highly Annoyed Population (HAP) and Highly Sleep Disturbed Population (HSDP) were reduced by 63–69% and 37–62%, respectively. These findings provide a more balanced view of the environmental gains possible under achievable recycling conditions.”
2) Material delivery distance:
Thank you for your feedback regarding the delivery distances of construction materials. We acknowledge your concern about the assumed transport distances and would like to clarify our rationale with supporting references.
Material delivery distances can vary significantly depending on geographic, logistical, and sourcing contexts. For instance:
• In Europe, delivery distances for steel and other materials range from 0 to 1900 km [1].
• A study conducted in Iran reported a steel delivery distance of approximately 670 km [2].
• In the United States, delivery distances for steel and timber were found to be around 160 km, while sensitivity analyses in the same study considered ranges from 0 to 10,000 km to assess transport impact variability [3].
• In New Zealand, steel was reported to be imported by ship from South Korea, covering a distance of approximately 10,000 km [4]. Another study in New Zealand reported a domestic steel delivery distance of around 182 km [5].
• In the Australian context, previous studies have assumed timber and steel delivery distances of approximately 200 km and 500 km, respectively [6].
The assumed steel delivery distance of 800 km is a conservative yet realistic estimate based on Australian supply chains. In the Northern Territory, iron ore is sourced from Frances Creek (~200–250 km from Darwin) and the Warrego project (~470 km away), while McArthur River Mine is ~970 km southeast of Darwin. In Sydney, steel is typically sourced from the Pilbara region in Western Australia, involving ~3,500–4,000 km of combined rail and sea transport. These figures demonstrate the long transport distances required for steel in both remote and urban areas, supporting the validity of the assumed 800 km in this life cycle assessment.
3) Noise exposed people calculation.
Updated in the section 3.1.1 (page 13-14)
“In this study, the number of highly annoyed people (HAP) and highly sleep-deprived people (HSDP) was primarily estimated using a percentage-based method. This approach assumes a uniform average population density within the noise-affected area and is commonly used in early-stage life cycle assessments (LCA) when high-resolution population data are unavailable.
To assess the robustness of this method, a supplementary analysis was conducted using Geographic Information System (GIS)-based population estimates. After calculating the construction noise levels, QGIS software version 3.4.8 was used to spatially estimate the number of people exposed to noise. Population data were sourced from the WorldPop database and integrated into QGIS for spatial analysis and visualisation. As illustrated in Figure 4, two road segments were analysed—one located in Darwin and the other in New South Wales (NSW)—to represent low- and high-populated areas, respectively. The estimated average population density in these areas is approximately 17 persons/km in Darwin and 82 persons/km in NSW.
The comparison of results reveals notable differences between the two methods. For concrete floor systems, the estimated number of HAP in Darwin increased from 3,411 (percentage-based) to 11,473 (GIS-based), and in NSW, from 134,405 to 138,911. For timber floor systems, the HAP in Darwin increased from 2,906 to 4,043, while in NSW, it decreased significantly from 132,552 to 44,680 (Figure 7).
A similar variation was observed in HSDP estimates. Under the percentage-based method, HSDP in Darwin and NSW was estimated at 2,323 and 78,027, respectively. However, when using the GIS-based method for concrete floor construction, these figures increased to 8,953 for Darwin and 80,747 for NSW. For timber floor systems, GIS-based HSDP estimates were 2,771 in Darwin and 24,993 in NSW.
These differences are attributed to the increased accuracy of the GIS method, which accounts for actual spatial population distribution and excludes non-residential or unpopulated zones. In contrast, the uniform percentage method may overestimate or underestimate population exposure, particularly in geographically and demographically heterogeneous areas.
While the percentage-based method remains valid for consistent comparisons across scenarios and is aligned with practices in similar LCA-based noise assessments, the GIS-based results add depth and spatial accuracy to the findings. Therefore, this dual-method approach strengthens the reliability of the study and highlights the potential for GIS integration in future construction noise assessments.”
Figure 7: Noise exposed population variation due percentage- and GIS-base calculation
Reference:
[1] O. Iuorio, L. Napolano, L. Fiorino, and R. Landolfo, “The environmental impacts of an innovative modular lightweight steel system: The Elissa case,” J. Clean. Prod., vol. 238, Nov. 2019, doi: 10.1016/j.jclepro.2019.117905.
[2] A. Oladazimi, S. Mansour, and S. A. Hosseinijou, “Comparative life cycle assessment of steel and concrete construction frames: A case study of two residential buildings in Iran,” Buildings, vol. 10, no. 3, Mar. 2020, doi: 10.3390/buildings10030054.
[3] F. C. Rios, D. Grau, and W. K. Chong, “Reusing exterior wall framing systems: A cradle-to-cradle comparative life cycle assessment,” Waste Manag., vol. 94, pp. 120–135, 2019.
[4] K. Roy, A. A. Dani, H. Ichhpuni, Z. Fang, and J. B. P. Lim, “Improving Sustainability of Steel Roofs: Life Cycle Assessment of a Case Study Roof,” Appl. Sci., vol. 12, no. 12, Jun. 2022, doi: 10.3390/app12125943.
[5] J. Kerr, S. Rayburg, M. Neave, and J. Rodwell, “Comparative Analysis of the Global Warming Potential (GWP) of Structural Stone, Concrete and Steel Construction Materials,” Sustain., vol. 14, no. 15, Aug. 2022, doi: 10.3390/su14159019.
[6] R. Sultana, T. Khanam, A. Rashedi, and A. Rajabipour, “Integrating Noise into Life Cycle Assessment for Sustainable High-Rise Construction : A Comparative Study of Concrete , Timber , and Steel Frames in Australia,” vol. 11, pp. 1–18, 2025.

Round 2
Reviewer 3 Report (Previous Reviewer 2)
Comments and Suggestions for Authors
The title and objective ‘This study aims to develop a comprehensive and unified framework for integrating noise pollution into the Life Cycle Assessment (LCA) of concrete construction’ are not aligned with the methodology. Why is the wooden structure used?
It is recommended that the title be changed to match the content, including wood or a comparative analysis of concrete and wood structures.
Reference is made to tables A1... but it is not indicated that they are in the Appendix.
In line 149 there is a typo: ‘aterial’ should be ‘material’.
Author Response
1:
Comment: The title and objective ‘This study aims to develop a comprehensive and unified framework for integrating noise pollution into the Life Cycle Assessment (LCA) of concrete construction’ are not aligned with the methodology. Why is the wooden structure used?
It is recommended that the title be changed to match the content, including wood or a comparative analysis of concrete and wood structures.
Response: Many thanks for your comments. Updated title is “Integrating Noise Pollution into Life Cycle Assessment: A Comparative Framework for Concrete and Timber Floor Construction”
2:
Comment: Reference is made to tables A1... but it is not indicated that they are in the Appendix.
Response: Updated at page no 200, 206, 218, 226, 386, 468, 484,and 505.
3:
Comment: In line 149 there is a typo: ‘aterial’ should be ‘material’.
Response: Updated
Round 3
Reviewer 3 Report (Previous Reviewer 2)
Comments and Suggestions for Authors
Ok
This manuscript is a resubmission of an earlier submission. The following is a list of the peer review reports and author responses from that submission.
Round 1
Reviewer 1 Report
Comments and Suggestions for Authors
- The quality of some figures (e.g., fig. 1 and 2 ) must be improved and ensured with a high-resolution.
- All the signs in equations should be denoted. And the order of equations was wrong, e.g., no equation number 4.
- Some figure captions should be rechecked and improved. For example, figs. 4-9. There are some repetitions or errors.
- Fig. 4a and 4b should be merged into one figure caption.
- The scientific merits and innovation should be highlighted in abstract or conclusions.
- The conclusions should be concise point-to-point for better reading.
- There are few citations in the past three years. Some recent publications should be further literature reviewed and supplemented in Introduction.
Author Response
Comments and Suggestions from reviewer 1:
1.The quality of some figures (e.g., figs. 1 and 2) must be improved and ensured with high resolution.
Response to feedback 1: Figure 1 & 2 updated in page no 4 & 6.
2.All the signs in equations should be denoted. And the order of equations was wrong, e.g., no equation number 4.
Response to feedback 2: All equations are in order from 1-6 on pages 8-10. All the signs are denoted.
3.Some figure captions should be reviewed and revised for improvement. For example, figs. 4-9. There are some repetitions or errors.
Response to feedback 3: All Figure (4-9) captions have been improved. There is no repetition at his moment.
4.Fig. 4a and 4b should be merged into one figure caption.
Response to feedback 4: Figures 4a and 4b are merged as Figure 4 and named as “Figure 4: Two locations (Darwin and Sydney) in Australia selected for this case study.”
5.The scientific merits and innovation should be highlighted in abstract or conclusions.
Response to feedback 5: In the abstract a new sentence is added “this study establishes a foundation for integrating noise into LCA, supporting sustainable material choices, environmentally responsible construction, and health-centered policymaking, particularly in noise-sensitive urban development.”
6.The conclusions should be concise point-to-point for better reading.
Response to feedback 6: Updated in page number 17 and 18.
7.There are few citations in the past three years. Some recent publications should be further literature reviewed and supplemented in Introduction.
Response to feedback 7: Updated with recent publication citation (21,29, 31,32,42,68,70,71, and 72)
4.7 Please find the further comments for this manuscript as follows:
- This work aims to bridge this gap by proposing a comprehensive framework that integrates noise impacts into the LCA methodology, focusing specifically on concrete construction. However, the integration noise impacts into the LCA methodology has already investigated by some previous works according to the literature review. The main innovation on incorporation both traffic and construction noise into the LCA of concrete construction should be further explained.
Response to feedback 4.7.1: New paragraph added in introduction in page 3
“While traffic noise and construction machinery noise have been individually recognised for their environmental and health impacts, they are rarely assessed together in an integrated framework. In real-world construction scenarios, these noise sources co-occur, cumulatively affecting both nearby residents and on-site workers. Traffic noise, primarily generated during material transportation phases, contributes significantly to community annoyance and sleep disturbance, especially in urban areas with high population density. Meanwhile, machinery noise, dominant during on-site construction activities, poses serious occupational health risks, including hearing loss and cardiovascular disorders among workers. Ignoring either source leads to an incomplete and underestimated assessment of the total noise-related burden. Furthermore, existing environmental evaluation tools and policies tend to address these sources separately, resulting in fragmented and less effective mitigation strategies. By integrating both mobile (traffic) and stationary (machinery) noise within the LCA framework, this study offers a more comprehensive and accurate method for quantifying the environmental and health impacts of concrete construction across its entire lifecycle.”
- Some comparison analyses between the proposed method and others should be elaborated in results and discussion.
Response to feedback 4.7.2: Updated in section 3.4 named as result validation, page 16-17.
- The mid-point and the end-point impact assessments have been addressed in detail, However, the reasons or motivations for the selected assessments have not been provided or explained.
Response to feedback 4.7.3: It has been discussed just after the mid-point (3.1a), page 11
“In this study, midpoint impact assessment was selected to provide a detailed, category-specific understanding of environmental burdens associated with construction activities. Midpoint indicators are grouped into two main categories: (1) traditional environmental indicators—such as global warming, stratospheric ozone depletion, ionic radiation, ozone formation (human health and terrestrial ecosystem), fine particulate matter formation, terrestrial acidification, freshwater and marine eutrophication, ecotoxicity, toxicity, land use, resource scarcity, and water consumption—and (2) noise-specific health indicators, namely highly annoyed people (HAP) and highly sleep-deprived people (HSDP)[67]. The traditional indicators reflect impacts caused by emissions and resource consumption throughout the construction life cycle, such as raw material extraction, transportation, manufacturing, and energy use. These do not include noise emissions directly. In this study, however, noise impacts are integrated into the LCA framework through separate noise-specific health indicators (HAP and HSDP), thus addressing a critical limitation of conventional LCAs that overlook acoustic pollution. “
and end-point impact assessment part (3.2), page 14
“The ReCiPe 2016 Endpoint (H) impact assessment method was employed to evaluate the long-term consequences of construction activities across three key areas of protection: human health, ecosystems, and resources. Compared to midpoint indicators, endpoint assessments are particularly valuable because they translate complex environmental emissions into tangible, decision-relevant outcomes—such as years of life lost or species affected. This enables more effective communication of environmental trade-offs, particularly when public health concerns like noise exposure are considered. Traditional endpoint LCAs capture human health damage largely through emissions-based pathways (e.g., air pollution, toxicity), but they rarely quantify direct health burdens from noise exposure. By calculating Disability-Adjusted Life Years (DALYs) from both material-related emissions and noise exposure, this study pioneers an integrated endpoint analysis that more fully represents construction's total impact on human health. While endpoint methods involve greater uncertainty due to value-based assumptions, they are essential for policy-oriented and holistic life cycle assessments. Accordingly, the hierarchist perspective was selected, representing a balanced and widely accepted scientific viewpoint [68].”
- Some recent and appropriate references should be supplemented, especially for the related publications in the last three years.
Response to feedback 4.7.4: Updated with recent publication citation (21,29, 31,32,42,68,70,71, and 72)
- The figure quality must be improved for better reading.
Response to feedback 4.7.5: All figures (Fig 1 -9) quality has been improved.

Reviewer 2 Report
Comments and Suggestions for Authors
I appreciate the invitation to review the article “Towards Sustainable Construction: A Novel Framework for Integrating Concrete Work Noise Impact in BIM-Life Cycle Assessment Method” which addresses an important issue that requires a great deal of research development for its improvement.
After review of the article, the process planted in the methodology to address the problem is interesting, but the development has shortcomings on important issues that cause me to propose its rejection, which I discuss below:
- In general, the title and the document is based on the LCA, however, construction and demolition are addressed, but not operation or use, and therefore the life cycle is not complete.
- Regarding methodological issues, BIM is stated in the title but throughout the article it is not explicitly stated what BIM brings to the research. A mention is made that Revit is used, but nothing more.
- Values and assumptions are adopted for the analysis without justification, such as the raw material extraction distance for concrete, which is not indicated for wood with which it is compared. Or the assumption of using 100% recycled concrete, when this is not possible. Likewise, the values taken from the database are not indicated. The authors themselves state this as a limitation of the research. These issues completely condition the results obtained and therefore the conclusions reached. As an idea for the authors, I consider that it would have been necessary to present the initial data and justify their validity for the case of application, as well as to carry out a sensitivity analysis.
- The very limitation of obtaining the population also makes the results questionable, since there are methods, as they indicate, for obtaining them.
- The same happens with the non-linear variation of noise, the authors themselves indicate that it is only valid for small variations, but there is no justification in the article that the situation analyzed meets this criterion.
- In point 4 it is stated that “The ReCiPe 2016 method has been applied to assess the results of this research.”, however, this method and how it is applied in the evaluation and validation of the results has not been explained in the methodology.
Some more specific questions:
- In the introduction, after stating the gap that is intended to be addressed, part of the methodology is stated, so this would not be the place to state this issue.
- It is stated in the methodology that further explanation is given in appendix 1, however only bibliographic references are listed.
- An explanation of where formulas 6 and 7 are derived from is needed.
- It is indicated that data are taken from Ecoinvent 3.8, it would be necessary to indicate the values adopted and their dispersion, for an assessment of the final results.
- From the reading of the text it is not possible to understand the relationships between noise and aspects such as terrestrial acidification or scarcity of fossil resources, among others. This requires a much more detailed justification.
- In Figure 5 a normalization is applied, but how this is done is not indicated. The noise is in logarithmic scale.
- Figure 6 has no explanation of the acronyms, HAP, NWS and HAP, NSW are used, are they the same? It also appears in Figure 7. In this graph, and in the document, LCA is mentioned, but the exploitation phase is not analyzed.
- The hypothesis of recycling 100% of the concrete without the need for new extractions is the only one that is not feasible in the current development of concrete recycling for structures.
- I do not know the particularities of Australia, but in other areas this distance would not be competitive, and is usually a few tens of kilometers. In this regard, the distances considered for timber are not indicated.
- Given the importance of this factor and others associated with distance, and in order to be able to obtain generalizable conclusions and make the research extrapolable to other regions, it is necessary to include a sensitivity analysis of the initial data and the results obtained.
- On the basis of which the value of 100% is adopted in Figure 7.
- In point 4.2 it is indicated that the DALY value for concrete is 0.029 and for wood it is 0.00392 and others integrating the noise, how are these values obtained?
Author Response
Comments and Suggestions from Reviewer 2:
I appreciate the invitation to review the article “Towards Sustainable Construction: A Novel Framework for Integrating Concrete Work Noise Impact in BIM-Life Cycle Assessment Method” which addresses an important issue that requires a great deal of research development for its improvement.
After review of the article, the process planted in the methodology to address the problem is interesting, but the development has shortcomings on important issues that cause me to propose its rejection, which I discuss below:
- In general, the title and the document is based on the LCA, however, construction and demolition are addressed, but not operation or use, and therefore the life cycle is not complete.
Response to the feedback: In appendix section 4 (Table A3, page 26-27) indicates the material quantity that shows each stage of the floor's lifetime for 100 years.
- Regarding methodological issues, BIM is stated in the title but throughout the article it is not explicitly stated what BIM brings to the research. A mention is made that Revit is used, but nothing more.
Response: Updated in Section 2.2 (page 6-7)
“In the construction stage, Building Information Modeling (BIM) was employed to streamline and improve the accuracy of material estimation and lifecycle assessment inputs [44]. Using Autodesk Revit 2022, a 3D model of the building was developed based on 2D drawings. Project parameters were defined in detail, including material types, dimensions, and structural elements. BIM enabled the automated generation of a material takeoff schedule, providing a precise and consistent bill of quantities (BOQ) for concrete and timber floors (Tables A1 and A2). This data was essential for quantifying material flows, energy use, and transportation impacts across lifecycle stages. By leveraging BIM, the study ensured methodological consistency, minimised human error in quantity estimation, and strengthened the reproducibility of the LCA framework.”
- Values and assumptions are adopted for the analysis without justification, such as the raw material extraction distance for concrete, which is not indicated for wood with which it is compared. Or the assumption of using 100% recycled concrete, when this is not possible. Likewise, the values taken from the database are not indicated. The authors themselves state this as a limitation of the research. These issues completely condition the results obtained and therefore the conclusions reached. As an idea for the authors, I consider that it would have been necessary to present the initial data and justify their validity for the case of application, as well as to carry out a sensitivity analysis.
Response: Distance can vary from 25 to 1600 km (Sporchia et al., 2025)(Eslami et al., 2024)(Jolly et al., 2024)(Lui, 2005). In this research, an initial travel distance of 50km and 800 km is used for timber and concrete materials. Later, during reuse, the distance is reduced to 25km and 50 km.
I acknowledge that current concrete recycling rates typically range from 80% to 90%, making 100% recycling challenging in practice (Balasbaneh & Sher, 2024)(Eslami et al., 2024) However, I applied the 100% recycling scenario as a best-case theoretical model, similar to other studies aiming to explore maximum environmental benefits. This approach helps illustrate the full potential of recycling advancements and aligns with future policy goals, even if not fully realized in today’s structural concrete recycling practices.
A sensitivity analysis (section 3.3, page 15) is included where noise mitigation measure is taken.
- The very limitation of obtaining the population also makes the results questionable, since there are methods, as they indicate, for obtaining them.
Response: new paragraph added in page 13 “A separate analysis was conducted to assess population exposure using Geographic Information System (GIS) software. The results indicate that, for concrete construction activities, the number of highly annoyed individuals in the Darwin area is estimated at 5,365—an increase compared to the previously calculated exposure population of 3,411. In the case of timber floor construction, GIS-based analysis estimates 4,863 highly annoyed individuals, which is slightly higherr than the earlier population-based calculation of 2,906. However, for both locations—Darwin and New South Wales—the number of highly sleep-disturbed individuals remains relatively consistent, with GIS analysis estimating 80,747 affected persons, closely aligning with the earlier estimate of 78,027.”
- The same happens with the non-linear variation of noise, the authors themselves indicate that it is only valid for small variations, but there is no justification in the article that the situation analyzed meets this criterion.
Response: We thank the reviewer for this insightful comment. We have now clarified this justification in the revised manuscript in Appendix Section 2.2a (page 23-24)
- In point 4 it is stated that “The ReCiPe 2016 method has been applied to assess the results of this research.”, however, this method and how it is applied in the evaluation and validation of the results has not been explained in the methodology.
It has explained in 3.1a and 3.2 (page 11 and 14)
“In this study, midpoint impact assessment was selected to provide a detailed, category-specific understanding of environmental burdens associated with construction activities. Midpoint indicators are grouped into two main categories: (1) traditional environmental indicators—such as global warming, stratospheric ozone depletion, ionic radiation, ozone formation (human health and terrestrial ecosystem), fine particulate matter formation, terrestrial acidification, freshwater and marine eutrophication, ecotoxicity, toxicity, land use, resource scarcity, and water consumption—and (2) noise-specific health indicators, namely highly annoyed people (HAP) and highly sleep-deprived people (HSDP)[67]. The traditional indicators reflect impacts caused by emissions and resource consumption throughout the construction life cycle, such as raw material extraction, transportation, manufacturing, and energy use. These do not include noise emissions directly. In this study, however, noise impacts are integrated into the LCA framework through separate noise-specific health indicators (HAP and HSDP), thus addressing a critical limitation of conventional LCAs that overlook acoustic pollution.”
Some more specific questions:
- In the introduction, after stating the gap that is intended to be addressed, part of the methodology is stated, so this would not be the place to state this issue.
Response: Part of the methodology has been removed from the introduction and added to the methodology section (Page 7-8).
- It is stated in the methodology that further explanation is given in appendix 1, however only bibliographic references are listed.
Response: Appendix 1 is added before the bibliography (Page 19-24). (Earlier it was attached separately)
- An explanation of where formulas 6 and 7 are derived from is needed.
Response : Appendix section 1(Page 19-24)is included for this explanation
- It is indicated that data are taken from Ecoinvent 3.8, it would be necessary to indicate the values adopted and their dispersion, for an assessment of the final results.
Response: Appendix section 4 , Table A3(Page 26-27)
- From the reading of the text it is not possible to understand the relationships between noise and aspects such as terrestrial acidification or scarcity of fossil resources, among others. This requires a much more detailed justification.
Explained in section 3.1a (page 11)
“It is important to note that no direct mechanistic linkage exists between noise and some midpoint categories, like terrestrial acidification or fossil resource scarcity. These categories are influenced primarily by material and energy flows, not acoustic emissions. However, noise impacts are included in the same LCA framework to offer a complementary perspective, comprehensively addressing environmental degradation and human health effects. Thus, the study presents both groups of indicators to emphasise the multi-dimensional nature of construction sustainability assessment “
- In Figure 5 a normalization is applied, but how this is done is not indicated. The noise is in logarithmic scale.
Explained in section 3.1a (Page 11) & 3.1b (Page 13).
- Figure 6 has no explanation of the acronyms, HAP, NWS and HAP, NSW are used, are they the same? It also appears in Figure 7. In this graph, and in the document, LCA is mentioned, but the exploitation phase is not analyzed.
Response: Explained and updated the Figure 7.
- The hypothesis of recycling 100% of the concrete without the need for new extractions is the only one that is not feasible in the current development of concrete recycling for structures.
Response: Thank you for your valuable comment. I acknowledge that current concrete recycling rates typically range from 80% to 90%, making 100% recycling challenging in practice. However, I applied the 100% recycling scenario as a best-case theoretical model, similar to other studies aiming to explore maximum environmental benefits. This approach helps illustrate the full potential of recycling advancements and aligns with future policy goals, even if not fully realized in today’s structural concrete recycling practices.
- I do not know the particularities of Australia, but in other areas this distance would not be competitive, and is usually a few tens of kilometers. In this regard, the distances considered for timber are not indicated.
Response: Distance can vary from 25 to 1600 km (Sporchia et al., 2025)(Eslami et al., 2024)(Jolly et al., 2024)(Lui, 2005). In this research, an initial travel distance of 50km and 800 km is used for timber and concrete materials. Later, during reuse, the distance is reduced to 25km and 50 km.
- Given the importance of this factor and others associated with distance, and in order to be able to obtain generalizable conclusions and make the research extrapolable to other regions, it is necessary to include a sensitivity analysis of the initial data and the results obtained.
In Figure 6 & 7, material transportation distance reduced to 50km from 800km (Appendix A3, Page 26). The impact has been compared. In section 3.3 sensitivity analysis has adopted (page 15-16).
- On the basis of which the value of 100% is adopted in Figure 7.
Explained and updated 3.1 a and 3.1b (page 13)
“All results in Figure 7 are presented using normalized midpoint values, where 100% represents the highest environmental impact value for each category across all scenarios. This comparative approach allows for easy visualisation of relative performance. The normalization method follows standard LCA practice by converting raw impact data into dimensionless scores scaled against a reference value, as detailed in Table A7 (appendix section 4). This table provides the underlying characterisation results and normalisation factors applied to both environmental and noise-related midpoint indicators.”
- In point 4.2 it is indicated that the DALY value for concrete is 0.029 and for wood it is 0.00392 and others integrating the noise, how are these values obtained?
Response Updated at 3.2 (page 15)

Reviewer 3 Report
Comments and Suggestions for Authors
This paper presents a framework that integrates both traffic and construction noise into the lifecycle assessment of concrete structures. However, its readability is currently very poor. Substantial revisions are necessary before it can be considered for publication, as detailed in the following comments.
(1) Several chapters and sections are either missing or misnumbered. Currently, Section 2.1, Chapter 3, and Chapters 5 and 6 are absent. Please review and correct the document's structure.
(2) Similarly, Equation (4) is missing, as well as Tables A1–A5. Please review and ensure all referenced elements are included.
(3) It is strongly recommended to include a literature review after the Introduction to contextualise this paper within the existing literature.
(4) All figures should be placed immediately after the paragraph in which they are first introduced. All figures should be thoroughly explained to help the reader understand how they were generated and what insights they offer.
(5) All symbols used in an equation should be fully defined and explained for clarity.
Author Response
Comments and Suggestions from reviewer 3:
This paper presents a framework that integrates both traffic and construction noise into the lifecycle assessment of concrete structures. However, its readability is currently very poor. Substantial revisions are necessary before it can be considered for publication, as detailed in the following comments.
(1) Several chapters and sections are either missing or misnumbered. Currently, Section 2.1, Chapter 3, and Chapters 5 and 6 are absent. Please review and correct the document's structure.
Response: All the chapter and section are updated.
(2) Similarly, Equation (4) is missing, as well as Tables A1–A5. Please review and ensure all referenced elements are included.
Response: All equation are number serially. Table A1-A8 in included in Appendix, section 4, page 25-32.
(3) It is strongly recommended to include a literature review after the Introduction to contextualise this paper within the existing literature.
Response: Thank you for your valuable suggestion. In response, we have revised the Introduction section to incorporate a more structured and comprehensive literature review. To enhance clarity and flow, we introduced subsections within the Introduction, including a dedicated section on Recent Literature Review, followed by a clearly defined Research Gap and Aim. This restructuring allows for better contextualization of our study within existing research while maintaining coherence with the overall paper structure.
(4) All figures should be placed immediately after the paragraph in which they are first introduced. All figures should be thoroughly explained to help the reader understand how they were generated and what insights they offer.
Response: All figures are placed immediately after the paragraph. All figure are explained also.
(5) All symbols used in an equation should be fully defined and explained for clarity.
Response: All symbols are defined and explaine (page 8-10)
